# Gas isotope thermometry in the South Pole and Dome Fuji ice cores provides evidence for seasonal rectification of ice core gas records.

Jacob D. Morgan[1], Christo Buizert[2], Tyler J. Fudge[3], Kenji Kawamura[4,5,6], Jeffrey P. Severinghaus[1], Cathy M. Trudinger[7]

[1]Scripps Institution of Oceanography, University of California, San Diego, La Jolla, CA 92093, USA
[2]College of Earth Ocean and Atmospheric Sciences, Oregon State University, Corvallis, OR 97331, USA
[3]Department of Earth and Space Science, University of Washington, Seattle, WA 98195, USA
[4]National Institute of Polar Research, Tokyo 190-8518, Japan
[5]Department of Polar Science, The Graduate University of Advanced Studies (SOKENDAI), Tokyo 190-8518, Japan
[6]Japan Agency for Marine Science and Technology (JAMSTEC), Yokosuka 237-0061, Japan
[7]Climate Science Centre, CSIRO Oceans and Atmosphere, Aspendale, Victoria 3195, Australia

*Correspondence to*: Jacob D. Morgan (jdmorgan@ucsd.edu)

**Abstract**

Gas isotope thermometry using the isotopes of molecular nitrogen and argon has been used extensively to reconstruct past surface temperature change from Greenland ice cores. The gas isotope ratios $\delta^{15}N$ and $\delta^{40}Ar$ in the ice core are each set by the amount of gravitational and thermal fractionation in the firn. The gravitational component of fractionation is proportional to the firn thickness and the thermal component is proportional to the temperature difference between the top and bottom of the firn column, which can be related to surface temperature change. Compared to Greenland, Antarctic climate change is typically more gradual and smaller in magnitude, which results in smaller thermal fractionation signals that are harder to detect. This has hampered application of gas isotope thermometry to Antarctic ice cores.

Here, we present an analytical method for measuring $\delta^{15}N$ and $\delta^{40}Ar$ with a precision of 0.002‰ per atomic mass unit, a two-fold improvement on previous work. This allows us to reconstruct changes in firn thickness and temperature difference at South Pole between 30 and 5 kyr BP. We find that variability in firn thickness is controlled in part by changes in snow accumulation rate, which is, in turn, influenced strongly by the along-flowline topography upstream of the ice core site. Variability in our firn temperature difference record cannot be explained by annual-mean processes. We therefore propose that the ice core gas isotopes contain a seasonal bias due to rectification of seasonal signals in the upper firn. The strength of the rectification also appears to be linked to fluctuations in the upstream topography. As further evidence for the existence of rectification, we present new data from the Dome Fuji ice core that are also consistent with a seasonal bias throughout the Holocene.

Our findings have important implications for the interpretation of ice core gas records. For example, we show that the effects of upstream topography on ice core records can be significant at flank sites like South Pole—they are responsible for some of the largest signals in our record. Presumably upstream signals impact other flank-flow ice cores such as EDML,

Vostok, and EGRIP similarly. Additionally, future work is required to confirm the existence of seasonal rectification in polar firn, determine how spatially and temporally widespread rectifier effects are, and to incorporate the relevant physics into firn air models.

## 1    Introduction

Past surface temperatures are commonly inferred from ice cores using the water isotope composition of the ice ($\delta^{18}O_{ice}$), which requires a site-specific calibration of the proxy. Early studies calibrated $\delta^{18}O_{ice}$ using its modern-day spatial relationship with mean annual temperature near the ice core site, which was hypothesized to be identical to the relationship with temporal variations in site temperature (e.g. Jouzel et al., 1993). Subsequently, temporal calibrations have become possible for cores from Greenland and Antarctica thanks to the development of independent methods of temperature

reconstruction based on borehole thermometry (Cuffey et al., 2016; Dahl-Jensen et al., 1998; Buizert et al., 2021) or gas isotope measurements (Severinghaus and Brook, 1999; Huber et al., 2006; Kindler et al., 2014; Buizert et al., 2014; Orsi et al., 2014; Buizert et al., 2021). Calibrations using these methods have shown that the temporal relationship between gas isotope measurements and temperature can indeed differ significantly from the spatial calibration, and can also vary in time. Unfortunately, such independent temperature reconstructions are more challenging for East Antarctic ice cores for two main

reasons. First, the low snow accumulation rates at these sites means heat diffusion erases some of the thermal history of surface temperature change that borehole thermometry relies on. Second, the smaller, more gradual surface temperature fluctuations typical of Antarctic climate result in a lower signal-to-noise ratio for gas isotope thermometry. This has made it more difficult to evaluate the calibration and reliability of $\delta^{18}O_{ice}$ as a paleotemperature proxy in East Antarctica.

In this paper, we present the first Antarctic application of gas isotope thermometry with the precision necessary to detect

interpretable signals. We describe an improved analytical method for making measurements of the isotopic composition of molecular nitrogen ($N_2$) and argon (Ar) on a single ice core sample and present data from the South Pole Ice Core between 5 and 30 kyr BP. Our method yields a two-fold improvement in precision compared to previous work, meaning we can measure the isotope ratios with a reproducibility of ~0.002‰ per atomic mass unit. This allows us to use the isotope measurements to separate the gravitational and thermal components of diffusive fractionation in the firn column and thus

quantitatively reconstruct past temporal changes in the height of the diffusive column of firn air and the temperature difference across it. The analytical precision corresponds to an uncertainty of ~1 m and 0.3°C in firn thickness and temperature difference respectively. Our measurements span the last glacial period, the deglaciation, and the early Holocene, recording changes in climate and firn properties throughout this time.

This study is important as the most meaningful test yet of gas isotope thermometry in Antarctica. Wider application of

comparable, high-precision measurements would provide a benchmark for testing the ability of firn densification models to accurately simulate the thermal properties of the firn column across a wide range of past and present climate conditions and has the potential to improve past temperature reconstructions for the South Pole ice core and at other sites in East Antarctica.

## 2    Reconstructing firn properties

To reconstruct the firn air diffusive column height (DCH) and vertical temperature difference ($\Delta T_z$, the difference between temperature at the surface and the lock-in depth), we measure the isotopic composition of molecular nitrogen and argon ($\delta^{15}N$ of $N_2$ and $\delta^{40}Ar$ of Ar) in air extracted from ice core samples. All isotope ratios are expressed in delta notation relative to the modern atmosphere in units of per mil (‰).

Importantly, changes in the isotopic composition of atmospheric nitrogen and argon are negligible over the timescales relevant for most ice core studies ($<10^5$ years) (Mariotti, 1983; Sowers et al., 1989; Bender et al., 2008). Therefore, deviations of the ice core gas composition from the modern atmosphere must arise locally in the firn column. Gas transport in the firn is primarily by molecular diffusion and two processes dominate isotopic fractionation of air: gravitational and thermal fractionation.

In the first case, gravitational settling causes enrichment of the heavy isotopes and molecules at the base of the firn due to the lack of turbulent mixing of the air. The amount of enrichment is described by the barometric equation (Craig et al., 1988; Sowers et al., 1989; Schwander et al., 1993).

$$\delta_{grav} = \left( exp\left(\frac{\Delta m^{a/b} gz}{RT}\right) - 1 \right) \cdot 1000‰ \cong \frac{\Delta m^{a/b} gz}{RT} \cdot 1000‰ \tag{1}$$

Here, $\delta_{grav}$ is the isotopic deviation in units of ‰, $\Delta m$ is the mass difference between the isotope pair $a$ and $b$, $g$ is the gravitational acceleration, $z$ is the firn air diffusive column height, $R$ is the ideal gas constant, and $T$ is the average temperature of the firn column in Kelvin. It is often useful to make a linear approximation to the exponential (via the first order Taylor expansion), as shown in Eq. (1), which adds a relative error of less than 0.5% for the range of values considered here. Firn thickness depends on the balance between the rates of snow accumulation and densification with both low temperatures and high accumulation rates resulting in a large $z$. Because site temperature and accumulation rate are strongly and positively correlated in the climate system, variations in $z$ tend to be muted. Broadly, the spatial pattern across Antarctica is one of thicker firn columns in colder locations, suggesting a dominance of the temperature effect. However, in comparing last glacial maximum (LGM) and pre-industrial values of $z$ in central Antarctica we find thinner firn columns during the colder LGM (Landais et al. 2006), suggesting a dominance of the accumulation effect (Buizert, 2021).

Second, gas composition is fractionated by temperature gradients in the firn, with heavier isotopes and molecules concentrated at the cold end of the gradient by thermal diffusion fractionation (Severinghaus et al., 1998). The magnitude of the fractionation is given by:

$$\delta_{therm} = \Omega^{a/b} \Delta T_z \tag{2}$$

where $\Omega$ is the empirically measured thermal diffusion sensitivity of the isotope pair $a$ and $b$ and $\Delta T_z$ is the temperature difference between the top and bottom of the diffusive column of air. Positive values of $\Delta T_z$ and $\delta_{therm}$ correspond to the top of the firn column being warmer than the base. At South Pole, the vertical temperature profile depends broadly on the balance between the downward advection of cold ice from the surface and the upward conduction of geothermal heat.

Perturbations to either the mean annual surface temperature, basal geothermal heat flux, ice thickness, or vertical velocities can all influence the firn temperature gradient. The height of the firn column at the South Pole (~120 m) makes it particularly

well suited to recording thermal perturbations because the thermal relaxation time of the firn column scales with the square of the firn column height (Cuffey and Paterson, 2010).

By measuring two isotope ratios, we can mathematically solve for the two components of fractionation, allowing us to calculate the height of the past diffusive column of air and the vertical temperature difference across it. To do so, we first use Eq. (1) and (2) to express $\delta^{15}N$ and $\delta^{40}Ar$ as the sum of their respective gravitational and thermal components:

$$\delta^{15}N = \frac{gz}{RT} + \Omega^{15/14}\Delta T_z \tag{3}$$

$$\delta^{40}Ar = 4\frac{gz}{RT} + \Omega^{40/36}\Delta T_z \tag{4}$$

Severinghaus et al. (2003) take advantage of the fact that (in the linear approximation) the gravitational fractionation term is four times larger for $\delta^{40}Ar$ than for $\delta^{15}N$ and define $\delta^{15}N_{excess}$, a second-order isotope parameter proportional to the temperature difference:

$$\delta^{15}N_{excess} = \delta^{15}N - \frac{1}{4}\delta^{40}Ar$$

$$= \left(\Omega^{15/14} - \frac{1}{4}\Omega^{40/36}\right)\Delta T_z \tag{5}$$

Similarly, we can define $\delta^{40}Ar_{DCH}$, a second-order isotope parameter directly proportional to the diffusive column height:

$$\delta^{40}Ar_{DCH} = \delta^{40}Ar - \frac{\Omega^{40/36}}{\Omega^{15/14}}\delta^{15}N$$

$$= \frac{gz}{RT}\left(4 - \frac{\Omega^{40/36}}{\Omega^{15/14}}\right) \tag{6}$$

The final step is to convert from the isotope parameters to $\Delta T_z$, the firn temperature difference, and $z$, the diffusive column

height by rearranging Eq. (5) and (6):

$$z = \frac{RT \cdot \delta^{40}Ar_{DCH}}{g\left(4 - \frac{\Omega^{40/36}}{\Omega^{15/14}}\right)} \tag{7}$$

$$\Delta T_z = \frac{\delta^{15}N_{excess}}{\Omega^{15/14} - \frac{1}{4}\Omega^{40/36}} \tag{8}$$

This conversion from isotope ratios to the firn physical properties assumes that the isotope ratios occluded in bubbles at the base of the firn column are in diffusive equilibrium with the local environment and that the only fractionating processes occurring are gravity and thermal gradients. This is generally true for the firn column at an ice core site, although we discuss in Sect. 5.2.4 reasons why this might not be the case at South Pole, Dome Fuji, and potentially other ice core sites.

## 3 Methods

### 3.1 Sample recovery and storage

The South Pole Ice Core SPC14 (hereafter SPICEcore) was drilled between 2014 and 2016 at a site close to the Amundsen-Scott South Pole Station (Casey et al., 2014; Winski et al., 2019; Epifanio et al., 2020; Souney et al., 2021). Ice cores were transported to the National Science Foundation's Ice Core Facility (NSF-ICF) in Denver, Colorado where 200g samples were cut and shipped to Scripps Institution of Oceanography in La Jolla, California. The ice was kept colder than -25°C from coring to analysis.

### 3.2 Gas extraction and mass spectrometry

Our method for the extraction and purification of the trapped gases is similar to that described by Kobashi et al. (2008) and Orsi (2013). Briefly, an 80 g piece of ice is melted in an evacuated vessel and the gases are stirred out of solution by a magnetic stir bar. Oxygen is removed by reaction with copper turnings heated to 500°C to prevent interference of the $^{18}O^{18}O$ isotopologue with the $^{36}Ar$ beam and to improve $^{29}N_2$ beam stability. Other interfering gases, such as water vapour and carbon dioxide, are removed by a series of glass u-traps immersed in liquid nitrogen at 77 K, and the remaining nitrogen and argon is cryogenically trapped in a stainless-steel tube immersed in liquid helium at 4 K. The dip tube is removed from the liquid helium and allowed to thaw and re-equilibrate for a minimum of 12 hours before being analysed.

Isotopic ratios of nitrogen ($^{29}N_2/^{28}N_2$) and argon ($^{40}Ar/^{36}Ar$) as well as the argon to nitrogen ratio ($^{40}Ar/^{28}N_2$) of the sample gas are measured on a dual inlet Thermo Finnigan MAT252 mass spectrometer. Routine laboratory corrections for source pressure imbalance and the $Ar/N_2$ chemical slope are made. Isotope and elemental ratios are expressed in units of ‰ relative to the modern atmosphere, sampled in La Jolla, California, USA. The La Jolla air samples are processed similarly to ice samples, meaning that small biases induced by gas handling cancel out to first order.

We make two important modifications to the methods described by Kobashi et al. (2008) and Orsi (2013). The first is a chemical slope correction to $\delta^{15}N$, which is artefactually enriched by the presence of $H_2$ in the sample gas. The second is the inclusion of a 30-minute delay between admission of the sample and reference gas into the bellows and the beginning of the measurement sequence. This is necessary due to an initial measurement bias caused by cooling of the bellows during expansion of the reference gas, which is at a higher pressure than the sample gas prior to expansion. Both modifications are discussed in more detail in Sect. S1.

### 3.3 Firn densification modelling

In this work we perform firn densification model simulations using a coupled firn densification-heat transport model that has been described previously elsewhere (Buizert et al., 2014, 2021). The model uses Herron-Langway densification physics formulated in terms of overburden pressure to allow for non-steady-state conditions (Eq. 4c in Herron and Langway, 1980). Firn thermal conductivity is based on Calonne et al. (2019) and other firn and ice thermal properties are based on Cuffey and

Paterson (2010). The forward model is forced using the surface temperature and accumulation rate histories at the site. The model simulates the time evolution of firn density and temperature with depth. The close-off density is estimated using the parameterization of Martinerie et al. (1994). Ice core gas properties (gravitational and thermal fractionation and gas age-ice age difference) are calculated and saved at the lock-in density, which is determined using the established approach by Blunier and Schwander (2000) of finding the lock-in density by subtracting a constant value from the Martinerie close-off density. Blunier and Schwander recommend a constant value of 14 kg m$^{-3}$ at Summit, Greenland. We use a value of 15 kg m$^{-3}$ based on modern-day observations at South Pole. The larger value reflects the fact that South Pole has a very thick lock-in zone. The DCH is equal to the lock-in depth minus the convective zone thickness; thermal fractionation is calculated using the temperature difference between the bottom of the convective zone and the lock-in depth. The convective zone thickness is set to 6 m and the firn surface density at 380 kg m$^{-3}$ following observations (Sowers, T. A. and Buizert, C., personal communication, 2021). Ice thickness and geothermal heat flux are held constant at 2600 m and 56 Wm$^{-2}$ respectively—these values are not very well known as SPICEcore was not drilled to bedrock. The model can be run in an inverse mode, in which an automated algorithm is used to estimate the temperature and accumulation histories that best fit the observational $\delta^{15}N$ data and the empirically reconstructed estimates of the gas age-ice age difference (Epifanio et al., 2020). We will refer to the optimized inverse scenario as the reference (REF) run; we later describe various model experiments that deviate from the REF scenario.

## 4    Results

We analysed samples from 170 depths in SPICEcore between 490 and 1310 m depth. The samples encompass bubble ice, clathrate ice, and the transition zone, where bubbles and clathrates coexist. We measured 14 depths in duplicate, giving us an estimate of analytical reproducibility. Our samples cover the time period from approximately 5,000 to 30,000 yr BP at an average resolution of 150 yr on the SP19 gas chronology (Epifanio et al., 2020). The measurements were made in two periods, between January and April 2018 and between October and December 2018. We calculate gravitationally-corrected $\delta Ar/N_2$ ($\delta Ar/N_{2\ grav}$) by making the common assumption that the enrichment per mass unit is equal to the measured $\delta^{15}N$ value (Craig et al., 1988; Bender et al., 1995) (thermal fractionation is negligible compared to the precision of the $\delta Ar/N_2$ measurement). We also make a small gas loss correction to $\delta^{40}Ar$ based on $\delta Ar/N_{2\ grav}$, the details of which are described in Sect. S2.

## 4.1 Reproducibility

We assess the reproducibility of our data by calculating the pooled standard deviation, $s_{pooled}$, which allows us to combine our replicate measurements and evaluate their deviations from their respective means:

$$s_{pooled} = \left( \frac{\sum_{j=1}^{m} \sum_{i=1}^{n_j} \left( \delta_{i,j} - \overline{\delta_j} \right)^2}{\sum_{j=1}^{m} n_j - m} \right)^{\frac{1}{2}} \tag{9}$$

where $\delta_{i,j}$ is the ith delta value for a replicate sample from the jth depth, $\overline{\delta_j}$ is the mean for all replicate samples for a given depth, $n_j$ is the number of samples analysed for a given depth and $m$ is the number of depth means analysed.

Five separate flasks of La Jolla air were analysed between 5 and 11 times each with at least one flask measured at the start and end of each measurement period. The pooled standard deviation of $\delta^{15}N$, $\delta^{40}Ar$, $\delta Ar/N_{2\ grav}$, and $\delta^{15}N_{excess}$ for the 40 total La Jolla air measurements is shown in Table 1. We achieve a two- and three-fold improvement, relative to Kobashi et al. (2008) and Orsi (2013), for ice measurements of $\delta^{15}N$ and $\delta^{15}N_{excess}$ respectively. We also note smaller improvements in the reproducibility of the other measurements and that some of the improvement may be due to superior ice quality for SPICEcore. This advance in measurement precision makes it possible to reliably detect the $\delta^{15}N_{excess}$ record of climatic signals in Antarctic ice for the first time.

**Table 1.** Mass normalised pooled standard deviation of replicate measurements of $\delta^{15}N$, $\delta^{40}Ar$, $\delta Ar/N_{2\ grav}$, and $\delta^{15}N_{excess}$ from either reference gas runs (REF), La Jolla air flasks (LJA), South Pole ice core samples (SPC) or other ice core samples. Units for all four isotope ratios are ‰ amu$^{-1}$ and the mass differences are 1, 4, 12, and 1 amu respectively. The final column indicates $n$, the number of samples used in the calculation.

| | $\delta^{15}N$ | $\delta^{40}Ar$ | $\delta Ar/N_{2\ grav}$ | $\delta^{15}N_{excess}$ | Num. Replicates |
|---|---|---|---|---|---|
| **This Study REF** | 0.0020 | 0.0023 | 0.0080 | 0.0023 | 58 |
| **This Study LJA** | 0.0027 | 0.0024 | 0.0042 | 0.0019 | 40 |
| **This Study SPC** | 0.0022 | 0.0030 | 0.0432 | 0.0013 | 14 |
| **Orsi LJA** | 0.003 | 0.0025 | 0.0073 | | 10 |
| **Orsi Ice** | 0.005 | 0.0036 | 0.0331 | 0.0042 | 169 |
| **Kobashi LJA** | 0.004 | 0.0035 | 0.0114 | | 17 |
| **Kobashi Ice** | 0.004 | 0.0040 | 0.0442 | | 148 |

It is also noteworthy that the mass-normalized pooled standard deviation of $\delta^{15}N_{excess}$ is smaller than that of $\delta^{15}N$ and $\delta^{40}Ar$ for the LJA and SPC samples. This suggests that the measured isotope ratios contain some mass-dependent variability that cancels out when we calculate $\delta^{15}N_{excess}$. The reproducibility of the reference gas samples does not show the same pattern, suggesting that the variability is introduced to the LJA samples during gas extraction rather than the mass spectrometry. For the SPC samples, another possibility is that the pattern is caused by real mass-dependent variability in the ice due to well-documented spatial heterogeneity in the depth of bubble close-off on a horizontal length-scale of a few

centimetres, i.e., similar to the width of an ice core sample (Orsi, 2013). This highlights the importance of measuring $\delta^{15}N$ and $\delta^{40}Ar$ on the same piece of ice. If $\delta^{15}N$ and $\delta^{40}Ar$ were measured on different pieces of ice, even adjacent pieces from the same depth in the core, this variability would not cancel out and would increase the scatter in $\delta^{15}N_{excess}$.

Finally, we note that the pooled standard deviation of $\delta Ar/N_{2\,grav}$ is much worse for the ice samples compared to the LJA measurements. This is because of similar cm-scale spatial heterogeneity in argon gas loss during bubble close-off and sample storage. Adjacent pieces of ice are likely to have lost different amounts of Ar so would not be expected to have the same $\delta Ar/N_{2\,grav}$ value.

## 4.2    Isotope and elemental ratios ($\delta^{15}N$, $\delta^{40}Ar$, and $\delta Ar/N_2$)

Our isotope ratio measurements are shown in Figure 1a. There is a strong positive correlation (r = 0.991) between $\delta^{15}N$ and $\delta^{40}Ar/4$ as the variability in both is dominated by gravitational fractionation, which affects mass-normalised isotope ratios equally. As described above, the difference between $\delta^{15}N$ and $\delta^{40}Ar/4$, termed $\delta^{15}N_{excess}$. reflects thermal fractionation in the firn column. $\delta^{15}N$ ranges from a minimum of 0.492‰ to a maximum of 0.626‰ with a mean of 0.562‰. $\delta^{40}Ar/4$ has a range of 0.497 to 0.625‰ with a mean of 0.569‰. Temporal variations are discussed below in the context of the firn properties calculated from the isotope ratios.

Gravitationally corrected $\delta Ar/N_2$ ($\delta Ar/N_{2\,grav}$) is depleted relative to the modern atmosphere over much of the depth range of our measurements, with values as low as -6.3‰. This is typical of ice core gas records and is due to preferential loss of Ar from the ice during bubble close-off (Craig et al., 1988; Bender, 2002; Severinghaus and Battle, 2006). However, there is also an interval of elevated $\delta Ar/N_{2\,grav}$ values between 8 and 18 kyr BP, with values as high as 7.8‰. This corresponds to the bubble-clathrate transition zone (BCTZ), where gas molecules are held in coexisting bubbles and clathrate hydrates. Here, $\delta Ar/N_{2\,grav}$ is enriched by the fact that post-coring gas loss occurs primarily from the bubbles, which are enriched in $N_2$ due to the stronger affinity of Ar for the clathrate phase (Bender et al., 1995; Ikeda-Fukazawa et al., 2001). The transformation to clathrates occurs heterogeneously throughout the core, increasing the scatter in our $\delta Ar/N_{2\,grav}$ measurements. Both the enrichment and increased scatter of elemental ratios in the BCTZ has been noted in many ice cores (Suwa and Bender, 2008a, b; Kobashi et al., 2008; Shackleton et al., 2019), but recent work appears to confirm that there is no appreciable isotope fractionation associated with clathration (Oyabu et al., 2021).

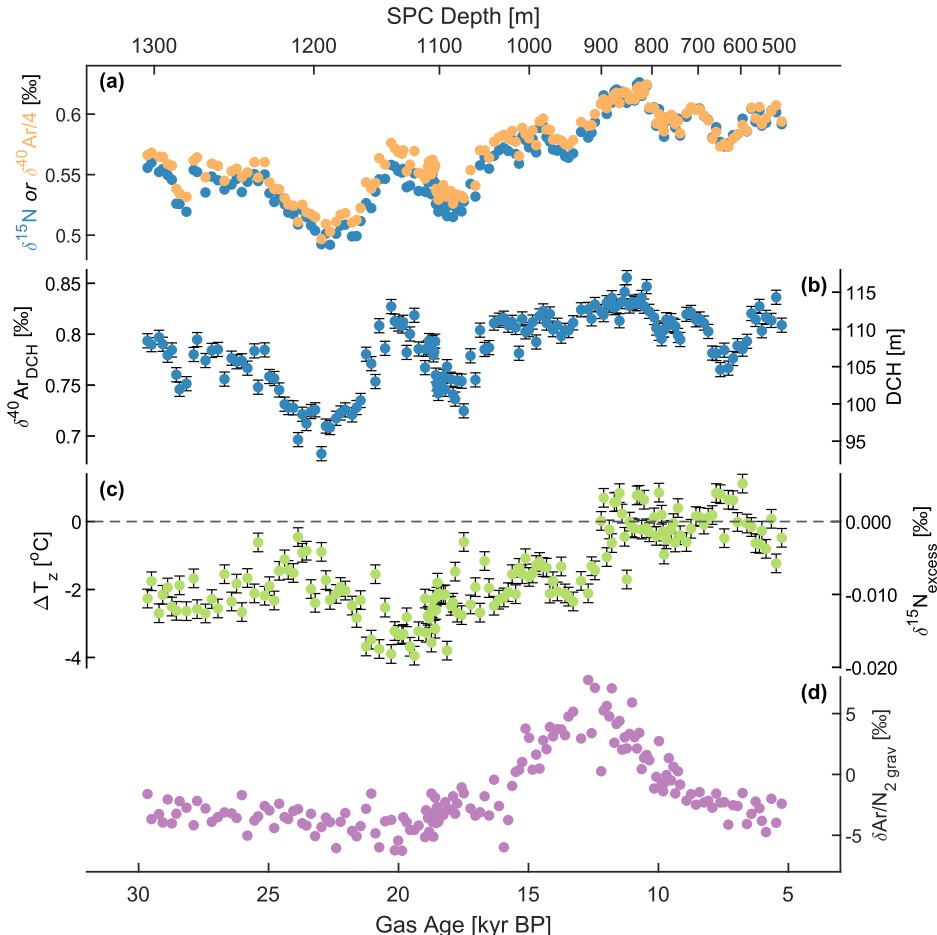

**Figure 1. (a)** SPICEcore measurements of $\delta^{15}N$ and $\delta^{40}Ar$. The $\delta^{40}Ar$ data are divided by 4 so that they can be plotted on the same axis as $\delta^{15}N$, with the visual offset between the two isotope ratios equal to $\delta^{15}N_{excess}$ (or, equivalently, $\Delta T_z$). **(b)** The firn diffusive column height (DCH) and **(c)** temperature difference ($\Delta T_z$) and equivalent isotope parameters (see Sect. 2 for explanation). **(d)** SPICEcore measurements of $\delta Ar/N_2$ after correction for gravitational fractionation. All data are plotted on the bottom x-axis on the SPC19 Gas Chronology. The corresponding depths in SPICEcore are indicated on the top x-axis. Error bars in DCH and $\Delta T_z$ represent one pooled standard deviation of replicate samples. Error bars for $\delta^{15}N$, $\delta^{40}Ar/4$, and $\delta Ar/N_{2\ grav}$ are smaller than the data markers.

## 4.3    Reconstructed firn properties

We calculate DCH and $\Delta T_z$ from our isotope data using Eq. (7) and (8). The time-series are shown in Figure 1. Both DCH and $\Delta T_z$ increase over the course of the record with DCH increasing from a glacial average of 103 m to 111 m in the Holocene, and $\Delta T_z$ increasing from -1.9°C to 0°C.

The minimum DCH we reconstruct is ~95 m and occurs around 23 kyr BP. The maximum DCH is ~115 m at ~11 kyr BP, although DCH is also >110 m around 20 kyr BP and for much of the deglaciation and early Holocene. The minimum (i.e., most negative) $\Delta T_z$ we reconstruct is -4°C, which occurs around 20 kyr BP, concurrently with a local maximum in firn

thickness. In fact, this inverse relationship is a persistent pattern on timescales of a few millennia. Despite a broad positive correlation between DCH and $\Delta T_z$ over the full 25 kyr record, a negative relationship exists between the higher-frequency

variability for much of the record (Figure 2). This is most evident prior to ~17 kyr BP, where the fluctuations in DCH and $\Delta T_z$ are the largest in amplitude and are clearly inverse of one another, but also exists in the younger part of the record (<12.5 kyr BP). For example, the most positive values of $\Delta T_z$ around 7 kyr BP are associated with a local minimum in DCH. The slope of the relationship is similar for glacial and Holocene samples, implying that the same physical process may be responsible. During the deglaciation, the inverse relationship between DCH and $\Delta T_z$ breaks down, with both properties

increasing through time. This time period is responsible for the overall positive correlation between the two time-series.

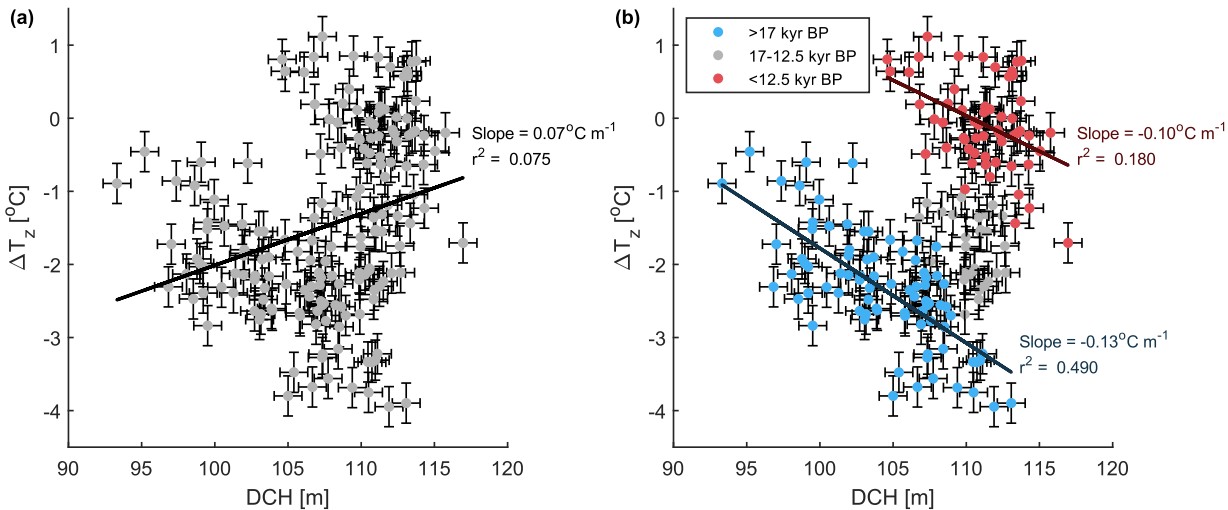

**Figure 2.** Firn temperature difference plotted against the diffusive column height. Error bars represent one pooled standard deviation of replicate samples. In panel **(a)**, the data are plotted in grey along with a regression to the entire dataset. In panel **(b)**, data older than 17 kyr are plotted in blue, data younger than 12.5 kyr are plotted in red, and a least squares linear fit to each subset is shown. The slope and squared correlation coefficient, $r^2$, of each fit is also indicated.

## 5    Discussion

The overall increase in both DCH and $\Delta T_z$ through the deglaciation is the expected response to increased snowfall due to a warming climate. DCH depends strongly on accumulation rate, which is higher in warmer interglacial periods than in glacials. For $\Delta T_z$, low accumulation in the glacial results in negative values as geothermal heat warms the base of the firn.

Then, as the accumulation rate increases, greater downward advection of cold surface ice makes the firn column closer to isothermal, making $\Delta T_z$ less negative. The concurrent increase in surface temperature during the deglaciation itself also acts to make $\Delta T_z$ more positive. Next, we discuss in more detail the processes responsible for the higher-frequency variability in DCH and $\Delta T_z$.

## 5.1 DCH (Diffusive Column Height)

First, we consider the mechanisms that drive changes in DCH. Winski et al. (2019) previously presented a record of SPICEcore $\delta^{15}N$, which, in the absence of complementary $\delta^{40}Ar$ measurements, they interpret solely as a gravitational fractionation/firn thickness signal. They argue, based on firn modelling experiments, that firn thickness variability in the Holocene is primarily controlled by the local accumulation rate. Our work supports this interpretation where we have data (11–5 kyr BP). Most of the variability in our $\delta^{15}N$ data is due to changes in gravitational fractionation and we note the correspondence between DCH and the SPICEcore record of accumulation between 11 and 5 kyr BP (Figure 3).

To further develop our understanding of the mechanisms driving changes in DCH, we must therefore consider the mechanisms that drive changes in the accumulation rate. In SPICEcore, Holocene accumulation rate variability is almost entirely explained by the spatial variability in accumulation upstream from the SPICEcore site (Lilien et al., 2018). This is because South Pole is located far from an ice divide, with ice flowing at 10 m yr$^{-1}$ in the direction of 40°W (Hamilton, 2004; Casey et al., 2014). Therefore, the snow deposition site for SPICEcore ice is further upstream for deeper, older ice. In this way, spatial accumulation variability is recorded as temporal variability in the ice core as more distant spatial anomalies are advected to a greater depth below the present-day SPICEcore site. The upstream spatial variability is in turn controlled directly by the local topography (Hamilton, 2004; Fudge et al., 2020). Namely, there is a positive corelation between the accumulation rate and the topographic curvature (second derivative along the direction of the flowline) (Figure 3, r = 0.55). The relationship is evident for at least 100 km in the upstream direction from South Pole and is consistent with findings at other sites in Antarctica (Waddington et al., 2007; Leonard et al., 2004) and Greenland (Miège et al., 2013; Hawley et al., 2014). The mechanism is that katabatic winds accelerate down slopes as the topography becomes steeper and decelerate as it becomes less steep. This results in greater erosion of snow from ridges (negative second derivative of elevation) and greater deposition in depressions (positive second derivative of elevation). In sum then, Holocene DCH is controlled in part by the upstream topography, via its dependence on the accumulation rate. This is most evident in our data between 8.5 and 6.5 kyr BP, where an ~8 m local minimum in DCH is co-located with a minimum in the modern spatial pattern of accumulation and with the steepest topographic slope upstream of SPICEcore.

The comparison between spatial (upstream) and temporal (SPICEcore) variability is less straightforward prior to 10 kyr BP because the exact position of the flowline is less certain and changes in climate are expected (Fudge et al., 2020). However, we hypothesize that the Holocene pattern also operated during the glacial period. For example, between 90 and 100 km upstream of South Pole, the topographic slope is close to or less than zero for the only extended period in the survey data (Figure 3). The survey line from Lilien et al. (2018) terminates at 100 km but data from the PolarGAP airborne radar campaign (Jordan et al., 2018) confirms that this feature is part of a broad topographic low on the flank of Titan Dome. The low is associated with a local maximum in both topographic curvature and upstream accumulation, suggesting that the topography does indeed cause higher accumulation in this region in the modern. Using a likely flowline, we find that ice of 20 kyr BP age would have originated at this topographic low. We therefore argue that the local maximum in DCH at 20 kyr

BP is due to greater net accumulation in the topographic low (Figure 3). Other Antarctic ice core records of $\delta^{15}N$ and accumulation rate show no evidence for continent-wide climatic changes at this time (Buizert et al., 2021), supporting our argument that this is a local signal, not a climatic one. Our argument requires certain features of the present-day topography to be unchanged over the past 25 kyr. This is certainly possible if these features are linked to the bedrock topography, as has been documented elsewhere in Antarctica (De Rydt et al., 2013).

Whilst some of the variability in DCH almost certainly reflects climatic changes associated with the deglaciation, it is not surprising that the effects of upstream variability are also present given the location of the SPICEcore site far from a dome. Our work shows that the signals associated with upstream effects can be substantial—the feature between 23 and 18.5 kyr BP is the largest in our record—and emphasises that caution must be applied when interpreting temporal changes in firn thickness in SPICEcore or other cores from flank sites such as EDML, Vostok, and EGRIP.

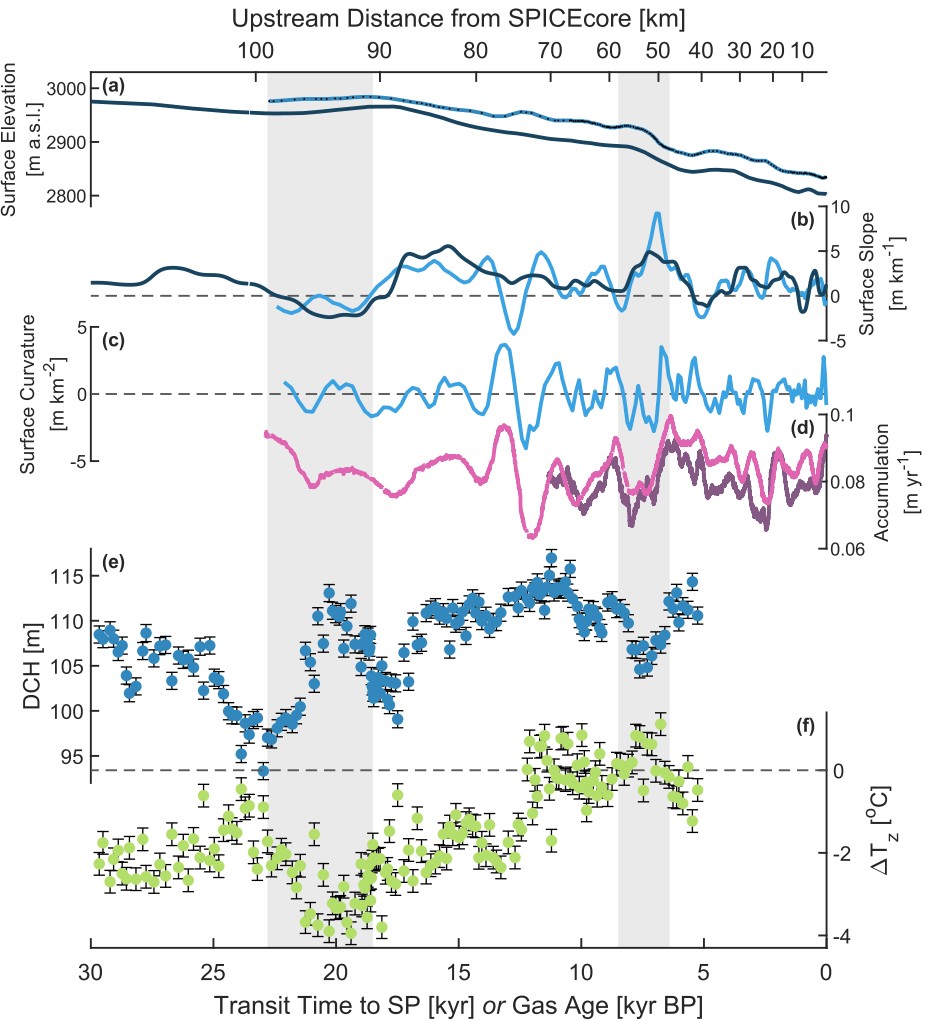

**Figure 3.** From top to bottom: **(a)** surface elevation profile (meters above sea level) along the flowline upstream of the SPICEcore site. Light blue line corresponds to snowmobile-mounted-GPS data from Lilien et al. (2018). Dark blue line corresponds to PolarGAP airborne radar data from Jordan et al. (2018). The gridded radar data were interpolated to the flowline using inverse distance weighted interpolation. **(b)** surface slope (first derivative) and **(c)** curvature (second derivative) of the elevation profiles, calculated in the direction of the flowline. Colours are as in (a). Curvature of the PolarGAP data is not shown due to the coarser spatial resolution of this dataset. **(d)** Pink line shows the accumulation rate along the flowline upstream of the SPICEcore site from Lilien et al. (2018). Purple line shows the SPICEcore accumulation rate derived from strain-corrected annual layer thicknesses (Winski et al., 2019) **(e)** firn diffusive column height and **(f)** temperature difference calculated from our isotope data. Upstream data in panels (a)–(d) are plotted on the bottom x-axis as functions of the transit time to the SPICEcore site based on the "mean" scenario in Fudge et al. (2020). The corresponding distances are shown on the top x-axis. SPICEcore data in panels (d)–(f) are plotted as functions of age on the SPC19 chronology. Grey shading highlights the changes in DCH and $\Delta T_z$ between 23 and 18.5 kyr BP and between 8.5 and 6.5 kyr BP, which are discussed in the text.

## 5.2   $\Delta T_z$ (Top-minus-bottom Firn Temperature Difference)

The variability in our record of $\Delta T_z$ is initially challenging to explain. We would have anticipated a positive correlation between DCH and $\Delta T_z$ since an increase in the accumulation rate ought to result in a thicker firn column and a weaker influence of geothermal heat on the temperature at the lock-in depth. However, although DCH and $\Delta T_z$ both increase over the course of the deglaciation, we instead observe a negative correlation between DCH and $\Delta T_z$ throughout most of the record (Figure 2). There must be some other mechanism that links variability in $\Delta T_z$ to either changes in accumulation or the local topography.

Furthermore, the -4°C difference in temperature between the top and bottom of the firn column is much larger in magnitude than the present-day temperature difference at South Pole, which is approximately 0°C (accumulation = 8 cm yr$^{-1}$). Other sites on the East Antarctic plateau with present-day accumulation rates comparable to the estimated glacial value at South Pole (4 cm yr$^{-1}$) also have smaller values of $\Delta T_z$, for example, -0.8°C at Dome C (2.5 cm yr$^{-1}$) and -1.3°C at Dome Fuji (2.5 cm yr$^{-1}$). Details of how present-day values of $\Delta T_z$ were determined are described in Sect. S3. Below, we examine several mechanisms that could explain the extreme negative values of $\Delta T_z$ and its inverse relationship with DCH in SPICEcore.

### 5.2.1   Surface temperature change

First, we investigate whether the SPICEcore $\Delta T_z$ reconstruction can be explained by variations in surface temperature. Changes in mean annual site temperature affect the firn temperature difference as the surface snow warms or cools and the vertical temperature profile in the ice sheet adjusts to a new equilibrium. Surface temperature change might also explain the negative relationship between DCH and $\Delta T_z$ in our data, with a surface cooling trend typically resulting in a more negative $\Delta T_z$ and a thicker firn column via reduced densification rates (Herron and Langway, 1980). We estimate the surface temperature history using the dynamical firn densification–heat transport model (Sect. 3.3) in an inverse mode. Briefly, the model adjusts initial estimates of past surface temperature and accumulation rate to best fit proxy-based reconstructions of firn thickness and gas age–ice age difference (Δage). This REF model run is able to produce a good fit to the proxy-based estimates of firn thickness and Δage for SPICEcore (Buizert et al., 2021) and also agrees well with our estimates of DCH and $\Delta T_z$ for much of the Last Glacial and Holocene periods (Figure 4). However, the modelled $\Delta T_z$ from the REF run does not agree well with our $\Delta T_z$ reconstruction during the LGM and for much of the deglaciation. The model does not reproduce the most negative values of $\Delta T_z$ between 23 and 18.5 kyr BP, nor does it capture some of the most positive values between 8.5 and 6.5 kyr BP. The firn model is not capable of fitting the $\Delta T_z$ data while simultaneously fitting the observational DCH and Δage data.

To evaluate what temperature history would be required to fit the $\Delta T_z$ data, we perform an additional experiment in which the firn temperature is decoupled from the firn densification physics (we call this the DECOUPLE run). In the DECOUPLE run the firn densification rates are not calculated, but rather they are read out from a data file corresponding to

the densification rates from the REF experiments. Accumulation rates are likewise equal to those from the REF run. The inverse model is then tasked to reconstruct the surface temperature history that best fits the $\Delta T_z$ data. The DECOUPLE surface temperature history required to fit our $\Delta T_z$ data is show in Figure 4. Note that the DECOUPLE scenario is not internally consistent, as the firn densification rates are inconsistent with the temperature forcing used. The design of the DECOUPLE experiment simply allows us to control the thermal gradient in the firn column (and thus $\Delta T_z$), while simultaneously ensuring we use the correct firn thickness and rate of downward advection of ice and heat.

We focus our interpretation of the DECOUPLE simulation on the direction and timing of changes in the inferred temperature history, rather than the absolute values, as $\Delta T_z$ is more sensitive to changes in surface temperature than to the temperature itself. Also shown for comparison is the optimal temperature history from the REF run (Buizert et al., 2021) and a temperature history from Kahle et al. (2021) based on a calibration of $\delta^{18}O_{ice}$ using the SPICEcore $\Delta$age data and the diffusion length of water isotopes in the firn. Note that both temperature histories are partially constrained by the $\Delta$age data, so they are not wholly independent.

The DECOUPLE temperature history differs substantially from the other temperature estimates. It features a prolonged 5°C cooling between 23 and 20 kyr BP, followed by a rapid 5°C warming from 13 to 11 kyr BP. The cooling event in the decoupled temperature history happens at a time when the other estimates indicate either constant temperatures or a slight warming associated with the initiation of the deglaciation, whereas the decoupled history shows almost no warming until the deglacial temperature change is almost fully realised in the other estimates. The timing and sign of temperature changes in the decoupled temperature history also bear little resemblance to other Antarctic ice core temperature reconstructions (Buizert et al., 2021; Uemura et al., 2018; Cuffey et al., 2016; Stenni et al., 2011; Jouzel et al., 2007; Petit et al., 1999).

Based on both the poor agreement of the measured and modelled $\Delta T_z$ in the REF run and the poor agreement of the DECOUPLE temperature history with other reconstructions, we conclude that surface temperature change is unlikely to fully explain our record of $\Delta T_z$, particularly the most negative values at 20 kyr BP.

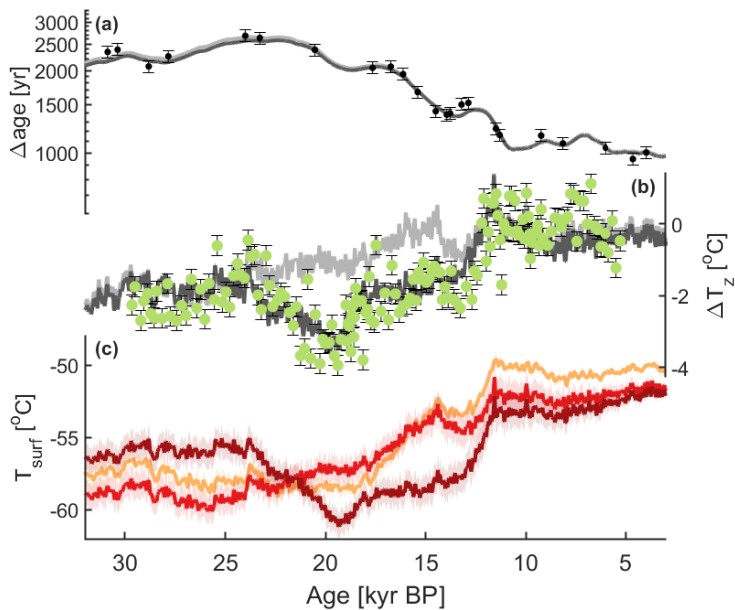

**Figure 4. (a)** Modelled Δage at SPICEcore in the REF (light grey) and DECOUPLE (dark grey) experiments. Markers show empirical Δage data from Epifanio et al. (2020) **(b)** SPICEcore $\Delta T_z$ reconstruction (green points) together with the modelled $\Delta T_z$ from the REF (light grey line) and DECOUPLE (dark grey line) model experiments. **(c)** South Pole surface temperature reconstructions from the REF (light red) and DECOUPLE (dark red) model experiments. Shading shows 1σ model uncertainty from Buizert et al. (2021). Also shown is a temperature reconstruction from Kahle et al. (2021) (orange), based on a calibration of $\delta^{18}O_{ice}$ using Δage and the water isotope diffusion length proxy. Shading shows 1σ standard deviation of model ensemble runs.

### 5.2.2 Ice thickness

Second, we investigate whether the SPICEcore $\Delta T_z$ reconstruction may be explained through variations in the thickness of the ice sheet. Ice thickness influences $\Delta T_z$ by controlling the vertical strain rate in the ice sheet, and thereby the downward advection of cold ice and the ability of geothermal heat to warm the base of the firn column. Temporal changes in ice thickness over the course of our record are certainly plausible, especially given that the flank location of South Pole means older ice originated upstream from the present-day SPICEcore site. The relevant parameter for our record of $\Delta T_z$ is the thickness of the ice column at the time and location that the bubbles were occluded. Variations in ice thickness experienced by SPICEcore ice are therefore the result of both temporal fluctuations in ice sheet elevation and upstream spatial fluctuations in ice thickness.

Temporal changes in ice sheet elevation at South Pole were estimated by Fudge et al. (2020) using output from a full ice-sheet model from Pollard et al. (2016). They conclude that changes in ice thickness have most likely been smaller than ±100 m in the past 20 kyr, with a mean elevation change of +16 m at 20 kyr BP. Certain combinations of model parameter values give changes between -325 and +250 m. Outside of the Holocene, the rate of change is always less than 20 m kyr⁻¹.

Spatial changes in ice thickness upstream of the SPICEcore site are dominated by changes in bed topography. For the Holocene, changes are well-known as the flowline is tightly constrained for this time period and the bed topography has been determined by ice penetrating radar (Lilien et al., 2018). Fluctuations are smaller than ±250 m. Beyond the Holocene, estimates of the bed topography do exist, but the exact flowline position becomes increasingly uncertain. Kahle et al. (2021) offer some constraints from their estimate of the SPICEcore thinning function and infer possible changes in bed elevation of around +200 m between 33 and 26 kyr BP, -200 m between 23 and 18.5 kyr BP.

We therefore seek to determine the maximum plausible change in $\Delta T_z$ by simulating a 500 m thickening or thinning using a using a 1-D ice flow model (Fudge et al., 2019). A 500 m change is 80 m larger than the range of present-day ice thickness fluctuations in the 100 km upstream of the SPICEcore site, corresponding to the past 20–25 kyr (Lilien et al., 2018); it is more than twice the magnitude of the changes in bed topography inferred by Kahle et al. (2021) in the past 30 kyr BP; and it is 1.5 to 2 times larger than the most extreme modelled surface elevation changes in the past 20 kyr. In our experiments, the modelled changes in ice thickness occur linearly over 2000 years and the post-change thickness is set to the present-day value of 2800 m in each case. We repeat each experiment twice with accumulation rates of 2 and 4 cm yr$^{-1}$.

The results of these simulations are shown in Figure 5. The thinning (thickening) experiments result in a decrease (increase) in $\Delta T_z$ to more (less) negative values. Lower accumulation rates correspond to more negative values of $\Delta T_z$, due to weaker downward advection of surface heat. The change in $\Delta T_z$ is less than 1°C for each experiment—3 times smaller than the largest change in our record. We therefore conclude that fluctuations in ice thickness are also unable to fully explain our observations.

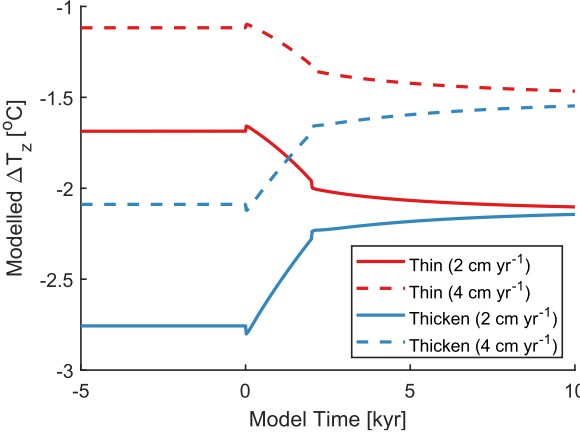

**Figure 5.** Modelled firn temperature difference for a thickening (blue lines) or thinning (red lines) of the ice column. Solid lines correspond to experiments with an accumulation rate of 2 cm yr$^{-1}$ and dashed lines correspond to an accumulation rate of 4 cm yr$^{-1}$.

### 5.2.3    Basal geothermal heat flux

Third, we investigate whether the SPICEcore $\Delta T_z$ reconstruction may be explained through variations in the basal geothermal heat flux (GHF). As discussed above, most East Antarctic ice cores have negative values of $\Delta T_z$, caused by

geothermal heat impinging on the base of the firn column due to low accumulation rates at these sites. Although the GHF at South Pole is constrained by borehole temperature measurements (Price et al., 2002; Beem et al., 2018) the firn temperature gradient at 20 kyr BP may have been set by very different basal conditions as the ice sheet flowed over regions of greater or lesser GHF towards the present-day SPICEcore site. In fact, a recent survey upstream of the SPICEcore site inferred values as high as 120 W m$^{-2}$ due to local faulting and hydrothermal activity (Jordan et al., 2018), more than double previous estimates for the region from continent-scale models (Van Liefferinge and Pattyn, 2013).

To test the hypothesis that the most negative values of $\Delta T_z$ and the negative relationship between DCH and $\Delta T_z$ are the result of spatiotemporal variations in GHF, we simulate the effect of a step change in the GHF using the firn densification model described in Sect. 3.3. To calculate an upper bound on the plausible change in $\Delta T_z$, we choose a low starting value of 40 W m$^{-2}$ and either double or triple the GHF instantaneously. We repeat the experiments with three difference ice thicknesses: 2800, 2300, and 1500 m. The first represents the present-day ice thickness at South Pole, and the second represents a plausibly thinner ice sheet (as deduced in Sect. 5.2.2). The third is a more extreme scenario, representing the minimum observed thickness in a recent survey of the upstream region (Beem et al., 2021). For all experiments, the accumulation rate is 4 cm yr$^{-1}$ and the surface temperature is -58°C, representing LGM conditions.

The results of the model experiments are shown in Figure 6. Over the course of ~100 kyr, the firn temperature profile adjusts to the new steady state, with $\Delta T_z$ decreasing by 0.7–2.9°C, depending on the ice thickness and GHF change. However, in our record of $\Delta T_z$, variability of this magnitude occurs in ~5 kyr. The model indicates that GHF changes can explain <0.5°C of change in $\Delta T_z$ on this timescale and only under the most extreme case of a tripling of GHF with a 1500 m ice column. Furthermore, a larger GHF would likely result in a slightly smaller DCH as the warmer firn column would densify more rapidly. This prediction is confirmed by the model (not shown) and is opposite to the inverse relationship we observe between DCH and $\Delta T_z$. Therefore, we conclude that spatial variability in the GHF upstream of South Pole is unable to fully explain our observations, particularly the rapid 3°C changes in $\Delta T_z$ between 23 and 18.5 kyr BP.

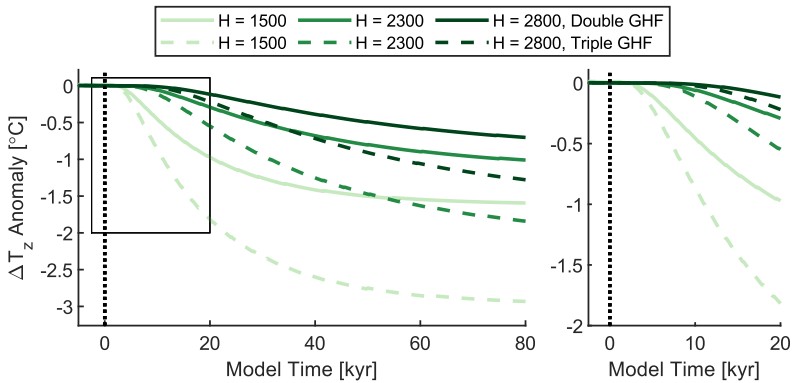

**Figure 6.** Modelled change in $\Delta T_z$ in response to a change in the basal geothermal heat flux (GHF), plotted as the difference from the value at t = 0. Solid lines correspond to experiments in which the GHF is doubled from 40 to 80 W m$^{-2}$ and dashed lines correspond to experiments in which the GHF is tripled from 40 to 120 W m$^{-2}$. Each experiment is repeated for three different ice thicknesses: 1500, 2300, and 2800 m. The vertical black line indicates $t = 0$, the time at which the change in GHF occurs in the model. The box in the left panel indicates the area covered by the right panel.

### 5.2.4    Rectification of seasonal thermal signals

Last, we investigate whether the SPICEcore $\Delta T_z$ reconstruction may be explained through so-called rectifier effects. Based on the evidence presented in previous sections, we argue that none of the processes known to control the annual-mean firn temperature profile can adequately explain our $\Delta T_z$ observations in terms of their magnitude, rate of change, and inverse relationship with DCH. Notably, however, temperature differences much larger than -4°C do arise on sub-annual timescales within the upper 20 m of the firn, at South Pole and elsewhere, in response to the seasonal surface temperature cycle (Dalrymple et al., 1966; Severinghaus et al., 2001; Brandt and Warren, 1997; Town et al., 2008). The corresponding gas isotope thermal fractionation signals only penetrate to ~10–15 m depth (Severinghaus et al., 2001; Weiler et al., 2009) and are typically assumed to cancel out each year such that the deep firn (and thus the ice core gas archive) reflects the annual-mean. Although existing firn air data from multiple sites are largely consistent with this assumption, the data are often lower precision than our measurements and are unlikely to represent a complete picture of firn processes on the spatial and temporal scales captured by our ice core record. Therefore, because annual-mean processes are unable to explain our data, we now investigate the possibility that some values of $\Delta T_z$ in our SPICEcore record could be the result of isotope signals in the deep firn being biased towards a particular season at certain times in the past. During winter, cold surface ice overlays a warmer firn column producing negative $\Delta T_z$ values in the upper firn; and vice versa during summer. To explain the most negative values of $\Delta T_z$ between 23 and 18.5 kyr BP, we infer a wintertime bias that is either weaker or non-existent at other times in the record. In addition, we propose that the most positive values between 8.5 and 6.5 kyr BP may represent a summertime bias. One notable aspect of this mechanism is that its strength can change very quickly—firn air convection appears and disappears on seasonal timescales. This may help to explain the changes in $\Delta T_z$ we observe between 23 and 18.5 kyr BP that are either too large or too abrupt to be explained by the processes discussed in previous sections.

In the sections below, we discuss mechanisms that might produce a summer or winter bias and argue that this hypothesis can explain many features of our dataset, including the most negative values of $\Delta T_z$, the rate at which they develop, and the inverse relationship between $\Delta T_z$ and DCH.

**5.2.4.1    Wintertime bias due to Rayleigh-Bénard convection**

Seasonal isotopic thermal fractionation signals in the firn are typically overwritten by the opposite signal of the following season (Severinghaus et al., 2001; Weiler et al., 2009). One way a seasonal bias can develop in the deep firn is if one season's isotope signal is preferentially preserved by being advected down into the firn, below the depth to which the next season's diffusive isotope signal penetrates. This type of differential preservation of winter versus summer signals due to 425 covariation of gas transport and concentration has been called a "seasonal rectifier effect" in prior literature (Denning et al., 1995; Severinghaus et al., 2001, 2010; Dreyfus et al., 2010; Trudinger et al., 2020). We adopt this language here.

This type of rectification requires a slow, non-turbulent, downward movement of air that occurs during one season but not the other. A plausible driving mechanism is the snow temperature inversion that arises in winter. Because snow and firn are efficient emitters in the infrared band and are usually warmed from below, their temperature is often coldest at the 430 surface. This is especially true in winter when incoming solar radiation is reduced or even absent. The temperature inversion results in an unstable air density profile in the firn that can trigger buoyancy-driven Rayleigh-Bénard convection, thus advecting seasonal isotope signals deeper into the firn. In this section we discuss evidence for this type of air movement in snow and firn and investigate its ability to explain our SPICEcore gas isotope records.

Sturm and Johnson (1991) demonstrated that buoyancy-driven overturning occurs readily in sub-Arctic snow in Alaska. 435 By making hourly observations of the three-dimensional temperature field within the winter snowpack for three years, they were able to observe large horizontal temperature gradients within the snow that were initiated and maintained by columns of rising warm air and sinking cold air. This convection occurred almost continuously throughout two successive winters. There is also ample evidence for air circulation within snow and firn from Antarctica, particularly if vertical cracks allow for fast upward return flow (Giovinetto, 1963; Albert et al., 2004; Fahnestock et al., 2004; Courville et al., 2007; Severinghaus 440 et al., 2010). Unfortunately, direct observations of changes in firn air composition associated with convection are scant since firn air sampling happens almost exclusively in the summer. However, there are published data from a winter firn air sampling campaign at South Pole. In this case, the authors did indeed find that the peak wintertime isotope signal occurred deeper than their firn air model predicted and speculated that this could be due to downward transport of the isotope anomaly by slowly sinking air (Severinghaus et al., 2001). If correct, this would provide confirmation not only of wintertime 445 convection at South Pole, but also that thermal isotope signals can be carried down into the firn by convection without being destroyed by turbulent mixing.

To test their hypothesis, we compare their wintertime firn air measurements from South Pole with values predicted by firn air model runs with and without parameterised Rayleigh-Bénard convection (Figure 7). In the model run without convection, the gases diffuse towards gravitational and thermal equilibrium as they are slowly advected downwards with the 450 densifying firn and occluded in bubbles in the lock-in zone. Because the model is one-dimensional, it is not possible to

explicitly simulate a three-dimensional Rayleigh-Bénard convection cell. Instead, we model just the sinking core of a convection cell, which we parameterise as an 8 cm d$^{-1}$ downward transport of gas between 0 and 20 m. Between 20 and 25 m, the downward transport decays to zero, resulting in mass convergence that would be balanced in the real world by horizontal transport and a return flux of gas to the surface. This approach allows us to approximate how the gas isotopes respond to convection using a one-dimensional model. The model run with downward transport better agrees with the observed wintertime firn air isotope ratios, with the negative wintertime values occurring deeper in the firn than in the model run with no downward advection. The model and the data therefore support our hypothesis that convection can carry seasonal thermal isotope signals down into the firn.

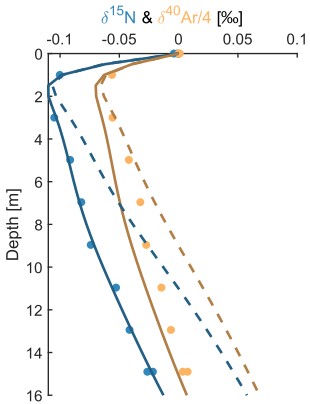

**Figure 7.** Shallow firn air data from the winter sampling campaign at South Pole by Severinghaus et al. (2001). Solid lines are a firn air model run that includes a slow downward advection of air between 0 and 25 m (see text for details). Dashed lines are the model run without any downward advection, as shown in Figure 5 in Severinghaus et al. (2001).

Because isotope data are only available in the top 16 m of the firn, we do not have an observational constraint on the strength of rectification in the deep firn, where ice core signals are recorded. To demonstrate that seasonal convection can affect isotope values in the deep firn, we perform an additional experiment with the firn air model. We simulate the isotope values in the full firn column under idealized South-Pole-like conditions (110 m thick firn, -51°C annual mean temperature, 7 cm a$^{-1}$ accumulation) and impose a 14 cm d$^{-1}$ downward advection throughout winter (April–September). In the model, the wintertime signal is advected deeper than the summer signal so is not fully cancelled out. This results in a -0.008‰ bias in the annual-mean signal in the deep firn compared to the control run with no downward advection (Figure 8, Movie S1). The bias is of comparable magnitude to the signals in our SPICEcore record, demonstrating that this mechanism could plausibly explain some of the millennial variability we observe.

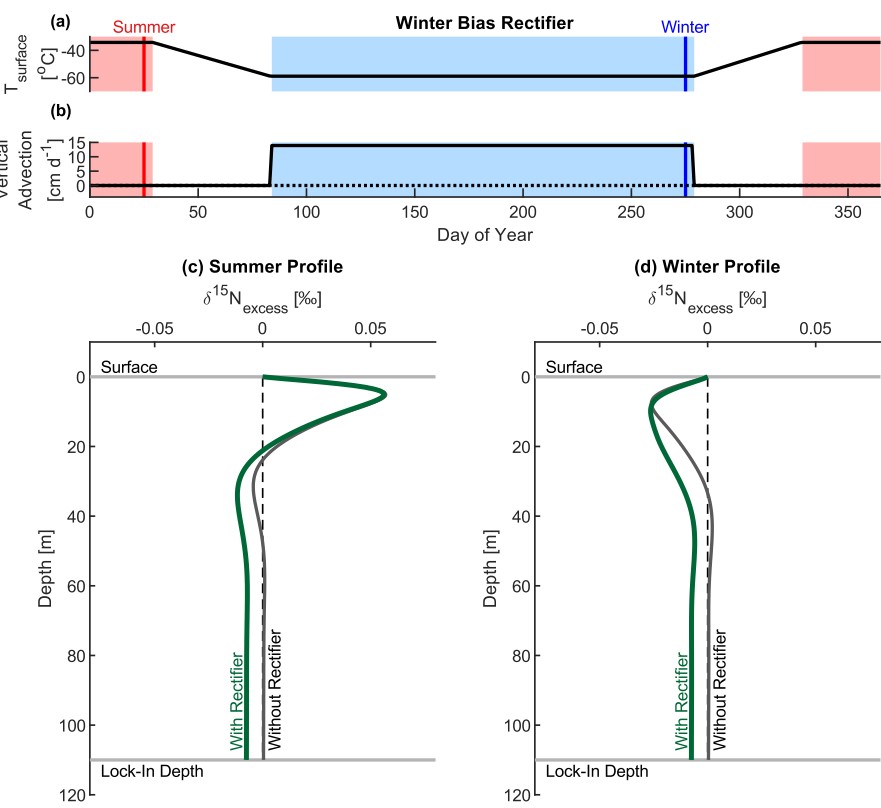

**Figure 8.** Results of idealized modelling experiment. Panels **(a)** and **(b)** show the temperature and advection forcing applied to the firn air model. The solid lines correspond to the "with rectifier" run and the dotted line in (b) corresponds to the "without rectifier" run with no vertical advection. Panels **(c)** and **(d)** show the vertical profile of $\delta^{15}N_{excess}$ in the firn column at the end of summer and winter respectively. The grey line is the run without advection, the green line is with advection. The days corresponding to the profiles are indicated by the vertical red and blue lines in the upper panels. An animated version of this figure is available in the supplement as Movie S1.

To explain the correlation between DCH and $\Delta T_z$ in our SPICEcore record, the strength of the wintertime convection must be linked to the wind speed and/or the topographic slope such that the rectifier is strongest when DCH is also at its
maximum. We hypothesize that this link is provided by the energy balance at the snow surface. Stronger katabatic winds on steeper slopes weaken the air temperature inversion by turbulently mixing heat down to the surface from aloft (Hudson and Brandt, 2005; Pietroni et al., 2014). The opposite is true in areas of minimal slope: weaker winds allow a strong inversion to develop via efficient loss of infrared radiation to space from the surface snow. This intense cooling of the surface promotes convection in the firn (Sturm and Johnson, 1991), which would strengthen the wintertime bias. Low wind speeds probably
also limit the formation of low permeability wind crusts that would inhibit convection (Domine et al., 2018). By this mechanism, the wintertime bias would be strongest and $\Delta T_z$ most negative in areas of flat topography, as we observe in our SPICEcore record (Figure 3).

As further evidence for this type of seasonal rectifier, we also present a previously unpublished $\Delta T_z$ record from the Dome Fuji ice core. The core was drilled in 1994-1996 and samples were stored at -50 °C until they were analysed at Scripps Institution of Oceanography in 2007 using a different method to our SPICEcore dataset (Bereiter et al., 2018). Briefly, an ice sample of 800–900 g was melted in an evacuated vessel, and the released air was continuously transferred to a dip tube through a -100°C water trap while stirring the melt water. The air sample was split in two aliquots (Method 1 in Bereiter et al., 2018), one was measured with Thermo Delta-Plus XP for $\delta^{15}N$ and the other was gettered to extract noble gases and then measured with Thermo Finnigan MAT252 for $\delta^{40}Ar$. The isotope data and the reconstructed $\Delta T_z$ data are shown in Figure 9, and we compare them to our estimate of the modelled Holocene $\Delta T_z$ from Buizert et al. (2021). The model estimate is based on the same firn densification modelling approach described in Sect. 3.3 constrained by Dome Fuji $\delta^{15}N$ and empirical $\Delta$age datasets described in Buizert et al. (2021). To estimate the uncertainty in the modelled $\Delta T_z$, we re-run the model with different values of the GHF and accumulation rate. We change the GHF by ±10 Wm$^{-2}$ and the accumulation rate by ±10%. The total uncertainty we report is the quadrature sum of the difference between these model runs and the optimal scenario.

Just like the SPICEcore record, the Dome Fuji $\Delta T_z$ data show evidence of a wintertime bias due to rectification. The mean of the Holocene $\Delta T_z$ data is more negative than both the present-day $\Delta T_z$ and the modelled Holocene $\Delta T_z$. Large changes in surface temperature, ice thickness, and GHF can be excluded during the Holocene, so we conclude that the mismatch is most likely due to rectification producing a wintertime bias throughout the Holocene at Dome Fuji. Because katabatic winds are weak at ice domes due to the flat topography, we expect that the wintertime Rayleigh-Bénard rectifier would be particularly effective at this site. This finding strengthens the case for the existence of rectification in Antarctica and demonstrates that rectification can affect gas records at both dome and flank sites and over a wide range of site characteristics (Dome Fuji is 1000 m higher in elevation, 5°C colder, and receives half as much snow accumulation).

Also plotted is the $\Delta T_z$ calculated from $\delta^{15}N$ and $\delta^{40}Ar$ measurements on firn air collected at Dome Fuji in 1998, which is -1.2°C (Figure 9, Sect. S3.2). This is more positive than the Holocene ice data and is consistent with the present-day observed firn temperature profile, suggesting no winter rectification is necessary to explain current conditions at Dome F. This could be due to cessation of rectification at some time during the past 2000 years, perhaps in the last century due to anthropogenic warming (the ice surface absorbs downwelling longwave radiation from greenhouse gases very effectively).

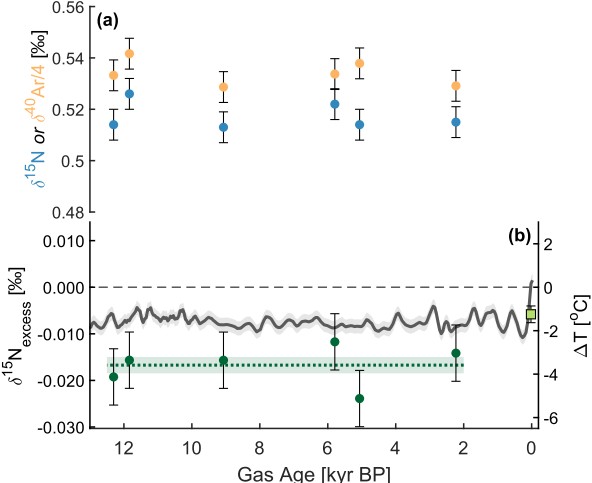

**Figure 9.** Measurements of **(a)** $\delta^{15}N$ and $\delta^{40}Ar$ used to calculate **(b)** $\delta^{15}N_{excess}$ and an estimate of $\Delta T_z$ from the Dome Fuji ice core. The $\Delta T_z$ data are plotted as dark green circles and compared to a model estimate of past $\Delta T_z$ at Dome Fuji from Buizert et al. (2021) (grey line and shading). The dashed green line shows the mean of the data and the shading represents one standard error of the mean of the six samples. The light green point shows an estimate of modern $\Delta T_z$ at Dome Fuji calculated using the method described in Sect. S3.2. The estimate is based on a new firn air dataset from archived samples collected in 1998 (Kawamura et al., 2006) and re-measured at SIO in 2008.

In summary, we propose that low wind speeds over areas of minimal topographic slope cause surface snow temperatures to be colder than on steeper slopes. In winter, this can result in an unstable air density profile in the firn and slow, non-turbulent convection of air to a depth of 10–20 m. This is deep enough to produce a cold, wintertime bias in our ice core records of $\Delta T_z$. In the Dome Fuji ice core, this bias existed throughout the Holocene until at least 2 kyr BP, whereas in SPICEcore, the cold bias is strongest at 20 kyr BP and is co-located with a thicker firn column due to the increased net accumulation of snow associated with slower and/or decelerating winds. Although this hypothesis is somewhat speculative, we believe this mechanism can plausibly explain (i) the most negative values in our record of $\Delta T_z$, (ii) the observed rate of change in $\Delta T_z$, and (iii) the inverse relationship with DCH.

#### 5.2.4.2 Summertime bias due to turbulent convective mixing

The slow, non-turbulent air circulation described above results in a wintertime bias in the deep firn. However, some sites in Antarctica experience vigorous turbulent mixing in the upper few meters of the firn column—termed the convective zone (Sowers et al., 1992; Bender et al., 1994; Kawamura et al., 2006; Severinghaus et al., 2010). This convective mixing of the free atmosphere into the surface firn "resets" the air composition back to atmospheric values, eroding the seasonal, gas-isotope thermal fractionation signals. The depth and extent of the mixing is controlled in part by the surface wind speed, with deeper convection associated with faster winds (Kawamura et al., 2006). Because katabatic winds are generally stronger in winter (van den Broeke and van Lipzig, 2003), we propose that a summer bias in $\Delta T_z$ could originate via a seasonality in the strength of convective mixing in the firn. We might expect stronger wintertime winds to be more effective than summertime

winds at eroding the thermal signals in the upper firn, meaning the summertime thermal signal would be preferentially preserved in the deep firn.

Again, we test the plausibility of this hypothesis with the firn air model. For this experiment, we parameterize the convective zone as an eddy diffusivity term in the upper 10 m of the firn that varies seasonally in magnitude–the eddy diffusivity is five times larger in winter than in summer. This dampens the wintertime thermal isotope signal meaning the

525 summer signal is preferentially preserved, resulting in a +0.008‰ bias in the annual-mean signal in the deep firn compared to the control run with no seasonal change in the eddy diffusivity (Figure 10, Movie S2). Again, the bias is of comparable magnitude to the signals in our SPICEcore record, demonstrating that this mechanism could plausibly explain some of the millennial variability we observe.

The summertime bias hypothesis is consistent with our SPICEcore data in that it predicts a deeper, stronger convective

zone on the steeper slopes ~50 km upstream of the SPICEcore site, where wintertime katabatic wind speeds would be faster (Vihma et al., 2011). This would produce a stronger, more positive bias in this location, potentially explaining the occurrence of positive values of $\Delta T_z$ (Figure 3). Although previous authors have speculated that this type of rectification could affect firn and ice core gas records (Severinghaus et al., 2001, 2010; Dreyfus et al., 2010; Petrenko et al., 2013; Verhulst, 2014), observational evidence is limited to one potential site (Law Dome, Antarctica; Trudinger et al., 2020).

Future firn air campaigns may help to uncover additional evidence of rectification via seasonal variability in convective strength.

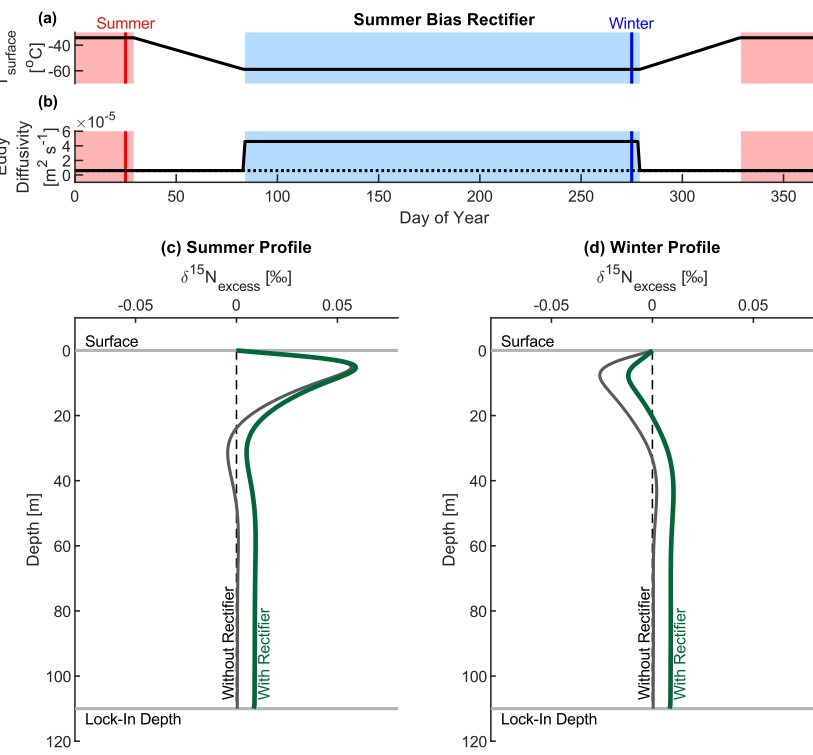

**Figure 10.** As in Figure 8, panel **(a)** shows the temperature forcing and panel **(b)** shows the eddy diffusivity forcing applied to the firn air model. The solid lines correspond to the "with rectifier" run and the dotted line in (b) corresponds to the "without rectifier" run with no seasonal change in eddy diffusivity. Panels **(c)** and **(d)** show the vertical profile of $\delta^{15}N_{excess}$ in the firn column at the end of summer and winter respectively. The grey line is the run without advection, the green line is with advection. The days corresponding to the profiles are indicated by the vertical lines in the upper panels. An animated version of this figure is available in the supplement as Movie S2.

## 6    Broader implications and future work

Our work demonstrates that gas isotope thermometry can provide meaningful paleoclimate information from Antarctic ice cores. The improved precision of our analytical method allows us to resolve changes in gravitational and thermal
fractionation throughout the last deglaciation although, for SPICEcore, the effects of upstream topography and possible seasonal rectification prevent us from making a surface temperature reconstruction. Application of measurements like ours to other ice cores is necessary to reveal how important these confounding factors are in other ice core gas records.

Rectification of ice core gas records has received limited attention in the literature so far, but our work argues that more careful consideration is necessary. Failure to recognise and account for rectifier effects where they do exist could potentially
lead to incorrect temperature estimates. Fortunately, it is unlikely that rectifier effects would have been significant for previous gas isotope thermometry studies in Greenland (e.g., Kobashi et al., 2007, 2011; Orsi et al., 2014; Landais et al., 2004, 2006; Huber et al., 2006). The presence of rectification via the mechanism we describe likely requires specific surface

conditions such as stagnant air and a strong atmospheric temperature inversion. These conditions probably occur rarely on the Antarctic plateau and are even less common in Greenland. To have any effect on the composition of air in the deep firn and closed-off ice they must persist for many weeks or months at a time and reoccur every year for many decades. Furthermore, in the case of Kobashi et al. (2011), agreement between their temperature reconstruction, regional climate model outputs and modern instrumental records also supports their analysis and interpretation. However, it might be necessary to account for rectifier effects in future gas isotope thermometry studies in Antarctica.

In principle, all gases would be affected by the processes we describe, not just nitrogen and argon. However, it is important to note that, if the seasonal bias we infer in $\Delta T_z$ is indeed thermal in origin, rectifier effects are likely smaller than typical signals of interest in many common ice core gas proxies. This is because the effect on isotopic and elemental ratios ought to be proportional to the thermal diffusion sensitivities of the gas pair. Thus, for a -3°C bias in $\Delta T_z$, the rectification of $CO_2$ concentration, for example, would be less than 0.3 ppmV (Weiler et al., 2009; Leuenberger and Lang, 2002). For $\delta^{15}N$, the bias is approximately 0.014‰ for each 1°C of rectification, corresponding to a 3 m bias in the calculated firn thickness. For gases with seasonal cycles in atmospheric abundance, the amount of rectification will be proportional to the amount of mixing or advection and the magnitude of the seasonal cycle, rather than thermal diffusion sensitivity. The signal size will therefore be specific to each site but is likely to be only a few percent of the seasonal variability (Trudinger et al., 2020).

In order to interpret ice core gas records accurately, including gas isotope thermometry data, it is crucial to determine the spatial and temporal prevalence of rectifier effects in Antarctica and Greenland and to learn more about the physical processes responsible. Important goals for future work would be to identify clear evidence for contemporary seasonal rectification in deep firn air and shallow ice samples and to determine the link to air transport in the firn and/or local meteorology. The topography upstream from South Pole would make a promising candidate site. It is possible that rectification will only affect sites with very specific conditions, meaning temperature reconstruction is a simpler task for other Antarctic cores. Alternatively, it may be possible to identify and correct for rectification effects using the isotope ratios of other inert gases such as Ne, Kr, and Xe (e.g. Kawamura et al., 2013). We also show that it is important to consider the effect of changes in basal geothermal heat flux and ice thickness when interpreting gas isotope thermometry data. The magnitudes of these effects are specific to each ice core site and should be considered when choosing candidate cores for gas isotope thermometry.

## 7    Conclusions

We present a new analytical method for measuring nitrogen and argon isotopes in ice core samples and the first major Antarctic application of gas isotope thermometry with the precision necessary to resolve typical Antarctic climatic signals. We quantitatively separate gravitational and thermal components of isotopic fractionation to reconstruct past changes in the height of, and temperature difference across, the diffusive firn column at South Pole. We find that both firn thickness and the firn temperature difference are influenced by local topographic variations along the flowline upstream from the ice core site.

The impact of topography generates the largest signals in our record, demonstrating that upstream effects must be considered when interpreting similar proxies in ice cores drilled at flank sites. At South Pole, firn thickness is greater in areas of negligible topographic slope due to greater net accumulation. Firn temperature gradient is also influenced by the topographic slope, potentially due to a seasonal rectification caused by the interaction of katabatic winds with surface topography and air in the uppermost firn column. Although we are unable to conclusively determine the origin of the rectifier, we suggest two

mechanisms that could plausibly be responsible. Similar evidence for rectification in the Dome Fuji ice core suggests that both dome and flank sites are susceptible. Until now, seasonal rectification has been assumed to have negligible impact on ice core gas records due to limited observational evidence. Our data shows that a more careful assessment of rectification is necessary to ensure accurate interpretation of gas isotope thermometry data from Antarctic ice cores and our new analytical technique can now be deployed to search for this effect at other sites. Determining how widespread rectification is, both

spatially and temporally, is crucial if gas isotope thermometry is to be used more widely in East Antarctica.

**Data availability**

The SPICEcore $\delta^{15}N$, $\delta^{40}Ar$, and $\delta Ar/N_2$ data associated with this study are archived online at the U.S. Antarctic Program Data Center (Morgan and Severinghaus, 2022) and are available from the corresponding author by request.

**Author contributions**

All authors contributed to this study. JDM made the SPICEcore ice core gas measurements with input from JPS. CB performed the firn densification modelling. TJF performed the ice flow modelling. KK made the DF firn air and ice core gas measurements. JDM, CB, TJF, KK, JPS, and CMT interpreted the results and contributed to the discussion. JDM wrote the paper with contributions from all authors.

**Competing interests**

The authors declare that they have no conflict of interest.

**Acknowledgements**

We thank the U.S. Ice Drilling Program for drilling the ice core; the 109th New York Air National Guard for airlift in Antarctica; NSF's Antarctic Infrastructure and Logistics and Antarctic Support Contractors, the SPICEcore field team, and the members of the South Pole station who facilitated the field operations; and the National Science Foundation Ice Core

Facility for ice core processing and archiving. We thank Ross Beaudette for help with the ice analysis, David Lilien, and Ed Brook for their helpful comments and discussion, and Benjamin Hills, Bob Hawley, and Max Stevens for sharing South Pole

firn temperature data. For the South Pole gas isotope measurements and firn densification modelling, we gratefully acknowledge funding from NSF (1443710, 1443472, and 1643394). KK and the Dome Fuji measurements were supported by JSPS and MEXT KAKENHI grants (18749002, 21671001, 15KK0027, and 17H06320). The manuscript was greatly improved thanks to constructive feedback from two anonymous reviewers.

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
