# Peer review of "Gas isotope thermometry in the South Pole and Dome Fuji ice cores provides evidence for seasonal rectification of ice core gas records."

_The Cryosphere, 2022_

## Referee Comment (RC1)

Review of Gas isotope thermometry in the South Pole and Dome Fuji ice cores provides evidence for seasonal rectification of ice core gas records." by Jacob D. Morgan et al, The Cryosphere.

**General:**

The manuscript presents new very interesting high-precision nitrogen and argon data for disentangling thermal from gravitational effects and come up with an improved interpretation of the vertical temperature gradients and firn thickness changes obtained from this measurement partitioning. The latter being partly influenced by snow accumulation, which is dependent on the topography along the ice flow line upstream at flank sites like South Pole. The authors state that observed temperature gradients in the firn cannot be explained by annual-mean processes alone and they therefore propose that there is a seasonal bias term present, rectifier effect, which strength itself is again dependent the topography upstream.

Major points:

Line 110ff: This conversion from isotope ratios to the firn physical properties assumes that the isotope ratios occluded in bubbles at the base of the firn column are in diffusive equilibrium with the local environment and that the only fractionating processes occurring are gravity and thermal gradients. This is generally true for the firn column at an ice core site, although we discuss in Sect. 5.2.4 reasons why this might not be the case at South Pole, Dome Fuji, and potentially other ice core sites.

What are the implications when the equilibrium will not be established? As the authors state the rectifier, effect is something that violates this assumption. How valid are then the results obtained in a first step assuming equilibrium conditions and then in a second step using this results and stating that there must be a rectifier effect at work. Is it somewhat a circular argument that can lead to such a statement, i.e. wrong assumption (equilibrium state reached) leads to a wrong partitioning of temperature gradients and firn thickness changes, which may then lead to a wrong interpretation ($\rightarrow$ rectifier). Please clarify that this is not the case.

Line 134ff: How and when is the hydrogen content measured?
Besides 28N2+ also 40Ar+ will react with H! This leads to a negative peak in 40Ar! Of course this happens also to 36Ar therefore the ratio 36/40Ar should remain rather stable in contrast to 29/28 where the mass/charge ratio decreases in contrast to 29 which loses 29N2 through the reaction with hydrogen but gains it from the same production using 28.

Have you looked into the stability of argon isotopes with varying hydrogen amounts in the sample?

Such reactions as mentioned are manifold in mass spectrometry. Have you looked into ArN2 formation and how it influences the isotopes of N2 and Ar?

4.1     Reproducibility: The authors did an excellent job in measuring the isotopic composition of nitrogen and argon with highest precision. Yet, there is still to be investigated, at least in my opinion, what kind of uncertainty is adequate to assign to a single depth measurement. In many publication so-called pooled standard deviation calculations have been used. Yet, this corresponds to a mean standard deviations based on replicated measurements on several depths. Whether this is an adequate measure is not clear to me. Maybe the authors can add some

argument why they think it is justified to use eq. 9. To be on the safe side one could argue to take the largest standard deviation of replicates or weight it according to the quality of the ice as bubble ice behaves differently than ice from the brittle zone or clathrate zone.

Table 1:   The fact that the pooled standard deviation of d15Nexcess for La Jolla air is lower than the d15N standard deviations shows that there is an instrument dependence present. Or is there an thermal diffusion fractionation expected during sampling of La Jolla air? I guess not since the inlet should be an aspirated intake ($\rightarrow$ R. Keeling publications).

Why is this standard deviation of d15N and d15Nexcess for ice core measurements smaller than for La Jolla air? Is it due to the lower number of samples?

Line 186ff: What about the possibility of instrumental influence that affect both nitrogen and argon isotopes? Can you exclude this? It would be worthwhile to report the reproducibility of standard gas admissions of both isotope ratios and report whether or not a co-variation exists. Furthermore, it would be good to mimic ice core measurements with aliquots of standard gas on bubble free ice.

Line 1990f: I agree if the assumption of a co-variation is true and not to be assigned to the instrument!

Minor points:

Line 146f: Can you explain why you choose a density correction of 15 kg/m$^3$?

Line 150f: Give a reference to this statement about the surface density

Line 156ff: This shortcoming is not directly addressed in the paper or do I miss something. Therefore, either skip this statement or add a statement how this shortcomings are addressed in the paper.

Line 255f: rewrite? The mechanism is that katabatic winds accelerate on steeper slopes and decelerate on less steep slopes.

Line 260f: add reference. Is this based on an ice sheet model study?

Figure 3: It is not clear from Fig. 3 which process is driving the DCH change (increase) in between the grey zones (i.e. from 19 kyr to 12kyr). The temperature is increasing. This should lead to a higher accumulation rate but this is not seen. Only a strong accumulation change is obvious between 14 and 13 kyr without a corresponding signal in DCH, only in $\Delta$Tz! Why?

Line 307ff: A similar study could be made with the uncertainty in the 40Ar measurements. What kind of uncertainty increase is necessary to be in agreement with the REF model output?

Line 424f: not clear? there must be a forward and backward movement possible. Need more explanation.

Line 451f: What means rapid. Here we talk about the enclosure time of several hundred years! If it changes from year to year, there must be a consistent regime over a very long time range at work to maintain this rectifier effect.

Line 484: This section Broader impact … could be combined with the conclusion section!

Supplementary material

Figure S1: Left panel, do the same for 40Ar!! It should remain constant, but it needs to be shown!!

Line 65f:  What kind of splining function was used?

Line 67ff:  This is very arbitrary

Figure S2: This is a very vague approach also indicated by a rather questionable slope calculation. What is the error of dAr/N2?

Would it not be more straight-forward to look at correlations of the d15Ntherm to d40Artherm reaching a slope as expected from laboratory thermal diffusion experiments, and approach it such that a loss of Ar is assumed following the measured Ar/N2 measurements.

Line 125: shallowest? Correct here, it corresponds to the deepest firn depths!

---

## Author Comment (AC1)

**Reply to Referee 1**

Thank you for your constructive and detailed review of the manuscript. Our response to your comments and the changes we plan to make to the manuscript are annotated below. The text from the review comments is in black italic text, our responses are in blue, and the changes we intend to make to the manuscript are in red.

Review of Gas isotope thermometry in the South Pole and Dome Fuji ice cores provides evidence for seasonal rectification of ice core gas records." by Jacob D. Morgan et al, The Cryosphere.

**General:**

The manuscript presents new very interesting high-precision nitrogen and argon data for disentangling thermal from gravitational effects and come up with an improved interpretation of the vertical temperature gradients and firn thickness changes obtained from this measurement partitioning. The latter being partly influenced by snow accumulation, which is dependent on the topography along the ice flow line upstream at flank sites like South Pole. The authors state that observed temperature gradients in the firn cannot be explained by annual-mean processes alone and they therefore propose that there is a seasonal bias term present, rectifier effect, which strength itself is again dependent the topography upstream. Maior points:

Line 110ff: This conversion from isotope ratios to the firn physical properties assumes that the isotope ratios occluded in bubbles at the base of the firn column are in diffusive equilibrium with the local environment and that the only fractionating processes occurring are gravity and thermal gradients. This is generally true for the firn column at an ice core site, although we discuss in Sect. 5.2.4 reasons why this might not be the case at South Pole, Dome Fuji, and potentially other ice core sites. What are the implications when the equilibrium will not be established? As the authors state the rectifier, effect is something that violates this assumption. How valid are then the results obtained in a first step assuming equilibrium conditions and then in a second step using this results and stating that there must be a rectifier effect at work. Is it somewhat a circular argument that can lead to such a statement, i.e. wrong assumption (equilibrium state reached) leads to a wrong partitioning of temperature gradients and firn thickness changes, which may then lead to a wrong interpretation ( $\rightarrow$  rectifier). Please clarify that this is not the case.

This is an excellent question worthy of a detailed answer. At the most basic level, the observation that we are unable to explain is that the difference between  $\delta^{15}N$  and  $\delta^{40}Ar/4$  is much larger than we would expect considering only the usual physical processes (gravity and thermal diffusion). This is true even without taking the step of calculating the firn thickness and temperature difference. From this primary observation we can conclude that some additional process is needed. In our opinion, a rectifier effect is the most likely candidate for this process because our investigation of changes in surface temperature, ice thickness, and geothermal heat flux show that thermal fractionation is not able to produce the difference in  $\delta^{15}N$  and  $\delta^{40}Ar/4$  that we observe. We are not aware of any other processes besides rectification that could explain the isotope data.

For the rest of the analysis, we rely on the assumption that the effect of the additional process on  $\delta^{15}$ N and  $\delta^{40}$ Ar scales in the same way as either gravitational or thermal fractionation. We consider this likely if the process is rectification and, provided this is true, it means our partitioning of  $\delta^{15}$ N and  $\delta^{40}$ Ar into their gravitational and thermal components is accurate, except one of the two components also contains the influence of the rectifier. Because we are unable to explain the  $\Delta T_z$  data via the typical processes that affect the firn temperature profile (surface temperature, ice thickness, geothermal gradient), and because

there is a plausible mechanism for seasonal rectification of thermal fractionation signal, it seems most likely that the additional process affects  $\Delta T_z$  rather than DCH.

It is possible that the additional process that enriches  $\delta^{40}$ Ar more than  $\delta^{15}$ N does not scale in the same way as either gravity or thermal fractionation. In this case, our partitioning would be erroneous and the DCH and  $\Delta T_z$  data are potentially somewhat misleading. However, this still would not result in a circular argument because rectification is still the best explanation for the enrichment of  $\delta^{40}$ Ar relative to  $\delta^{15}$ N.

Line 134ff: How and when is the hydrogen content measured? Besides 28N2+ also 40Ar+ will react with H! This leads to a negative peak in 40Ar! Of course this happens also to 36Ar therefore the ratio 36/40Ar should remain rather stable in contrast to 29/28 where the mass/charge ratio decreases in contrast to 29 which loses 29N2 through the reaction with hydrogen but gains it from the same production using 28. Have you looked into the stability of argon isotopes with varying hydrogen amounts in the sample? Such reactions as mentioned are manifold in mass spectrometry. Have you looked into ArN2 formation and how it influences the isotopes of N2 and Ar?

We measure the hydrogen concentration in the sample gas as a routine part of the sample analysis on the MAT252 mass spectrometer. We did indeed test to see if Ar isotopes would also be affected by processes such as the ones you describe. The results of our chemical slope experiments convinced us that there is no significant effect of H2 on  $\delta^{40}$ Ar at our level of precision. This is probably because the argon isotopes that we measure are separated by 4 mass units (36 and 40) so H36Ar+ does not interfere with the 40Ar+ beam, in contrast to H28N2+, which does interfere with the 28N2+ beam. We will add Ar isotope data to Fig. S1 that show constant Ar isotopes in our chemical slope experiments.

To find the value of the chemical slope, we add increasing amounts of pure H2 to aliquots of a reference gas with a well-known isotopic composition and measuring the resulting mass 29 enrichment. There is no significant change in  $\delta^{40}$ Ar as the H2 beam intensity increases, likely because H36Ar+ does not interfere with the 40Ar+ beam.

**Figure S1.** (a) Results of a H2 chemical slope experiment showing the increase in  $\delta^{15}$ N associated with an imbalance in the H2 concentration in the sample and reference aliquots. There is no detectable enrichment in  $\delta^{40}$ Ar. The slope and squared correlation coefficient of each least squares linear fit is also indicated. Uncertainty in the value for each aliquot is smaller than the data markers. (b) Measurements of  $\delta^{15}$ N from single sample-reference integration cycles (red and blue points), plus a 16-cycle running mean, showing the heavy bias immediately after expansion of the reference aliquot into the dual inlet bellows. Data from two replicate aliquots are shown, together with the average (horizontal black line) and standard deviation (grey shading) of all data later than 30 minutes after expansion (vertical dotted line).

4.1 Reproducibility: The authors did an excellent job in measuring the isotopic composition of nitrogen and argon with highest precision. Yet, there is still to be investigated, at least in my opinion, what kind of uncertainty is adequate to assign to a single depth measurement. In many publication so-called pooled standard deviation calculations have been used. Yet, this corresponds to a mean standard deviations based on replicated measurements on several depths. Whether this is an adequate measure is not clear to me. Maybe the authors can add some argument why they think it is justified to use eq. 9. To be on the safe side one could argue to take the largest standard deviation of replicates or weight it according to the quality of the ice as bubble ice behaves differently than ice from the brittle zone or clathrate zone.

For this study, we were more severely limited by sample size due to the smaller diameter of SPICEcore relative to previous cores. This made it very difficult to make as many duplicate measurements as we would have liked. However, we worked hard to make sure we were able to analyze 14 duplicate samples to give us some estimate of the reproducibility. Because many of the data points are single samples, we are not able to use the standard deviation of replicate measurements as an estimate of the uncertainty for each data point. Instead, we seek to use some statistically representative measure of the most likely estimate of the uncertainty. In our opinion, the pooled standard deviation is the best metric for this. It also makes for ready comparison with previous studies, as in Table 1. We feel that using the largest deviation of replicates as an estimate of the error would be overly conservative because the true uncertainty in most of the data points would be much smaller. Assigning different error estimates to the bubble ice, BCTZ ice, and clathrate ice is an excellent idea and would work well for a dataset with a larger number of replicate measurements. However, we feel unable to do so here as our estimate of the error would be based on fewer than 5 replicate measurements for each ice type. In sum, we believe that using the average standard deviation of each set of replicates from their respective mean (i.e., the pooled standard deviation) is the best choice for this work.

Table 1: The fact that the pooled standard deviation of d15Nexcess for La Jolla air is lower than the d15N standard deviations shows that there is an instrument dependence present. Or is there an thermal diffusion fractionation expected during sampling of La Jolla air? I guess not since the inlet should be an aspirated intake ( $\rightarrow R$ . Keeling publications).

See response and modified table below

Why is this standard deviation of d15N and d15Nexcess for ice core measurements smaller than for La Jolla air? Is it due to the lower number of samples?

Yes, quite possibly. Alternatively, it could be due to a small amount of error being added during sample handling. Although we aim to treat aliquots of La Jolla Air and ice core air as similarly as possible, there are obviously some unavoidable differences in the sampling and handling of the gas.

Line 186ff: What about the possibility of instrumental influence that affect both nitrogen and argon isotopes? Can you exclude this? It would be worthwhile to report the reproducibility of standard gas admissions of both isotope ratios and report whether or not a co-variation exists. Furthermore, it would be good to mimic ice core measurements with aliquots of standard gas on bubble free ice.

Excellent point, thank you for bringing this to our attention. We will add a row to the table listing the pooled standard deviation of standard gas runs on the mass spectrometer. We will also normalize the numbers by the mass difference of the isotope pair to allow for easier comparison.

As you can see from the new table, the reproducibility of  $\delta^{15}N$ ,  $\delta^{40}Ar$ , and  $\delta^{15}N_{excess}$  from the standard gas runs are identical. This suggests that there is no instrumental influence. However, as you point out, the reproducibility of  $\delta^{15}N_{excess}$  is lower than  $\delta^{15}N$  and  $\delta^{40}Ar$  for LJA and SPC samples, suggesting that we introduce some additional (approximately) mass-dependent error during gas handling for LJA and SPC samples. Because it is mass-dependent, it cancels out when calculating  $\delta^{15}N_{excess}$  so the reproducibility of this parameter is lower. It is noteworthy that the improvement in the reproducibility of  $\delta^{15}N_{excess}$  compared to  $\delta^{15}N$  and  $\delta^{40}Ar$  is even greater for SPC samples than for LJA samples. You are correct that we cannot rule out the possibility that it is due to gas handling, although the mechanism we originally proposed likely also plays a role. We will rewrite the sentence to include both possibilities.

**Table 1.** Mass normalised pooled standard deviation of replicate measurements of  $\delta^{15}$ N,  $\delta^{40}$ Ar,  $\delta$ Ar/N2 grav, and  $\delta^{15}$ Nexcess from either reference gas runs (REF), La Jolla air flasks (LJA), South Pole ice core samples (SPC) or other ice core samples. Units for all four isotope ratios are % amu-1 and the mass differences are 1, 4, 12, and 1 amu respectively. The final column indicates *n*, the number of samples used in the calculation.

|                | $\delta^{15}N$ | δ 40 Ar | $\delta Ar/N_{2grav}$ | $\delta^{15}N_{excess}$ | Num. Replicates |
|----------------|----------------|--------------------|-----------------------|-------------------------|-----------------|
| This Study Ref | 0.0020         | 0.0023             | 0.0080                | 0.0023                  | 58              |
| This Study LJA | 0.0027         | 0.0024             | 0.0042                | 0.0019                  | 40              |
| This Study SPC | 0.0022         | 0.0030             | 0.0432                | 0.0013                  | 14              |
| Orsi LJA       | 0.003          | 0.0025             | 0.0073                |                         | 10              |
| Orsi Ice       | 0.005          | 0.0036             | 0.0331                | 0.0042                  | 169             |
| Kobashi LJA    | 0.004          | 0.0035             | 0.0114                |                         |                 |
| Kobashi Ice    | 0.004          | 0.0040             | 0.0442                |                         |                 |

It is also noteworthy that the mass-normalized pooled standard deviation of  $\delta^{15}N_{excess}$  is smaller than that of  $\delta^{15}N$  and  $\delta^{40}Ar$  for the LJA and SPC samples. This suggests that the data contain some massdependent variability that cancels out when we calculate  $\delta^{15}N_{excess}$ . The reproducibility of the reference gas samples does not show the same pattern, suggesting that the variability is introduced to the LJA samples during gas extraction rather than the mass spectrometry. For the SPC samples, another possibility is that the pattern is caused by real mass-dependent variability in the ice due to well-documented spatial heterogeneity in the depth of bubble close-off on a horizontal length-scale of a few centimetres, i.e., similar to the width of an ice core sample (Orsi, 2013). This highlights the importance of measuring  $\delta^{15}N$ and  $\delta^{40}Ar$  on the same piece of ice. If  $\delta^{15}N$  and  $\delta^{40}Ar$  were measured on different pieces of ice, even adjacent pieces from the same depth in the core, this variability would not cancel out and would increase the scatter in  $\delta^{15}N_{excess}$ .

Finally, we note that the pooled standard deviation of  $\delta Ar/N_{2 \text{ grav}}$  is much worse for the ice samples compared to the LJA measurements. This is because of similar cm-scale spatial heterogeneity in argon gas loss during bubble close-off and sample storage. Adjacent pieces of ice are likely to have lost different amounts of Ar so would not be expected to have the same  $\delta Ar/N_{2 \text{ grav}}$  value.

**Line 1990f: I agree if the assumption of a co-variation is true and not to be assigned to the instrument!**

**See response above.**

**Minor points:**

Line 146f: Can you explain why you choose a density correction of 15 kg/m3?

We will rewrite the sentence as below.

Ice core gas properties (gravitational and thermal fractionation and gas age-ice age difference) are calculated and saved at the lock-in density, which is determined using the established approach by Blunier

and Schwander (2000) of finding the lock-in density by subtracting a constant value from the Martinerie close-off density. Blunier and Schwander recommend a constant value of 14 kg m-3 at Summit, Greenland. In the modern-day observations at SP this value is 15 kg m-3.

**Line 150f: Give a reference to this statement about the surface density**

The surface density is based on unpublished density measurements made on shallow cores at the SPICEcore site. We will add a "personal communication" reference as below.

The convective zone thickness is set to 6 m and the firn surface density at 380 kg m-3 following observations (Sowers, T. A. and Buizert, C., personal communication, 2021).

Line 156ff: This shortcoming is not directly addressed in the paper or do I miss something. Therefore, either skip this statement or add a statement how this shortcomings are addressed in the paper.

You are correct that the model-data mismatch is not addressed in this paper. We will remove the paragraph below, which begins on line 156 and ends on line 161.

Previous work has suggested firn densification models may have difficulty simulating the firn thickness in East Antarctica during glacial periods. During these periods ice core  $\delta 15N$  data show a firn column that is thinner than at present, whereas early densification model results suggested a thicker glacial firn column (Landais et al., 2006). Proposed solutions to this model-data mismatch include hypothesized glacial firn softening by dust loading (Freitag et al., 2013), and a strong temperature dependence of the firn thermal activation energy (Bréant et al., 2017); neither of these solutions improves the model-data 160 agreement at all sites simultaneously, though.

**Line 255f: rewrite? The mechanism is that katabatic winds accelerate on steeper slopes and decelerate on less steep slopes.**

This sentence was confusingly written. Our aim is to explain the link between wind speed and the second derivative (i.e., slopes that are becoming steeper). We will rewrite as below. Hopefully you agree that this version is clearer.

The mechanism is that katabatic winds accelerate down slopes as the topography becomes steeper and decelerate as it becomes less steep.

**Line 260f: add reference. Is this based on an ice sheet model study?**

Added a reference to Fudge et al. (2020) at the end of the sentence beginning on line 262.

The comparison between spatial (upstream) and temporal (SPICEcore) variability is less straightforward prior to 10 kyr BP because the exact position of the flowline is less certain and changes in climate are expected (Fudge et al., 2020).

Figure 3: It is not clear from Fig. 3 which process is driving the DCH change (increase) in between the grey zones (i.e. from 19 kyr to 12kyr). The temperature is increasing. This should lead to a higher accumulation rate but this is not seen. Only a strong accumulation change is obvious between 14 and 13 kyr without a corresponding signal in DCH, only in  $\Delta$ Tz! Why?

We agree that it is surprising that the largest feature in the surface curvature and upstream accumulation time series does not show up in our record of DCH. We considered several possible explanations, all of which are somewhat speculative.

- 1) The effects of the topography were outweighed by climatic changes Between 19 and 12 kyr BP, we would expect that climatic changes due to the deglaciation would be much larger than outside this time period during the LGM and Holocene. Perhaps between 19 and 12 kyr BP, the climatic changes are the dominant effect on DCH and  $\Delta T_z$ . The positive correlation for the grey points in Fig 2(b) is consistent with this. Increasing temperatures during the deglaciation would increase both  $\Delta T_z$  and DCH due to an increase in the accumulation rate. This mechanism could explain the large increase in  $\Delta T_z$  at 14 kyr BP that you pointed out, which is associated with a more gradual increase in DCH.
- 2) The feature existed but the flowline did not pass over it. The past flowline is increasingly uncertain for older ice and at 14 kyr BP there is no constraint on past ice velocities from the modern-day accumulation pattern (see Lilien et al. (2018); Fudge et al. (2020). The absence of any large change in DCH at 14 kyr BP could be because the flowline actually passed a little to the north or south of the feature. We have no information about the cross-flowline extent of the feature, but it does not appear in the admittedly coarser resolution satellite dataset.
- 3) The surface curvature feature between 80 and 70 km upstream was not present at 14 kyr BP. If the feature only developed within the last 14 kyr, it would not have been around to affect the gas isotopes in the ice forming at that spot, at that time.

**Line 307ff: A similar study could be made with the uncertainty in the 40Ar measurements. What kind of uncertainty increase is necessary to be in agreement with the REF model output?**

The largest disagreement between the REF model output and the  $\Delta T_z$  reconstruction is -3°C at approx.. 20 kyr BP. This corresponds to a 0.014‰ difference in  $\delta^{15}N_{excess}$  and therefore a 0.014\*4 = 0.056‰ difference in  $\delta^{40}Ar$ . This is approximately 4 times larger than our 1 $\sigma$  analytical error of 0.012‰.

**Line 424f: not clear? there must be a forward and backward movement possible. Need more explanation.**

We expanded the explanation of the convection parameterization based on your comment and comments from the other referee. See below.

In the model run without convection, the gases diffuse towards gravitational and thermal equilibrium as they are slowly advected downwards with the densifying firn and occluded in bubbles in the lock-in zone. Because the model is one-dimensional, it is not possible to explicitly simulate a three-dimensional Rayleigh-Bénard convection cell. Instead, we model just the sinking core of a convection cell, which we parameterise as an 8 cm d-1 downward transport of gas between 0 and 20 m. Between 20 and 25 m, the downward transport decays to zero, resulting in mass convergence that would be balanced in the real world by horizontal transport and a return flux of gas to the surface. This approach allows us to approximate how the gas isotopes respond to convection using a one-dimensional model.

Line 451f: What means rapid. Here we talk about the enclosure time of several hundred years! If it changes from year to year, there must be a consistent regime over a very long time range at work to maintain this rectifier effect.

You are right. It is too vague to use the word "rapid" here. We will re-write as below.

This may help to explain the changes in  $\Delta T_z$  we observe between 23 and 18.5 kyr BP that are either too large or too abrupt to be explained by the other hypotheses discussed above

**Line 484: This section Broader impact ... could be combined with the conclusion section!**

We have further expanded this section based on comments from another referee. It now includes discussion of the implications of our findings for previous gas isotope thermometry studies. We feel it now warrants a separate section.

**Supplementary material**

Figure S1: Left panel, do the same for 40Ar!! It should remain constant, but it needs to be shown!!

Yes, I agree. We will add Ar isotopes to this figure. See also the response above.

**Line 65f: What kind of splining function was used?**

We chose to fit a cubic smoothing spline using the built-in MATLAB function fit(). This type of spline seeks to minimizes both a measure of the goodness of fit and the second derivative of the fitted curve, with the smoothing parameter controlling the trade-off between the two. We will re-write the sentence to make this clearer.

To isolate the gas loss fractionation signal in our time-series, we attempt to empirically capture and remove the shared climate variability of  $\delta^{15}$ N and  $\delta^{40}$ Ar by fitting a smoothing spline. The spline fit seeks to minimize both the misfit to the data and the second derivative (roughness) of the fitted curve. The smoothing parameter, S, controls the trade-off between a perfect fit to the data (S = 1) and a perfectly smooth curve (S = 0). We fit splines for a range of values.

**Line 67ff: This is very arbitrary**

Yes, we agree. The choice of smoothing parameter is inherently subjective. See response below for justification.

Figure S2: This is a very vague approach also indicated by a rather questionable slope calculation. What is the error of dAr/N2? Would it not be more straight-forward to look at correlations of the d15Ntherm to d40Artherm reaching a slope as expected from laboratory thermal diffusion experiments, and approach it such that a loss of Ar is assumed following the measured Ar/N2 measurements.

Your idea of trying to use  $\delta^{15}N_{therm}$  and  $\delta^{40}Ar_{therm}$  to constrain the gas loss sounds interesting, but I'm not sure exactly what you mean. Because we calculate the gravitational and thermal components using the laboratory-determined coefficients, the slope of  $\delta^{15}N_{therm}$  to  $\delta^{40}Ar_{therm}$  is always exactly equal to the ratio of  $\Omega^{15/14}$  and  $\Omega^{40/36}$ . I am open to trying it out if you are able to explain it in more detail. We accept that this approach is somewhat speculative, but we believe that our approach is justified empirically by the results of the spline fitting and residual analysis. If we had only  $\delta^{40}Ar$  data, it would be impossible to know whether the correlation we observe between the spline residuals were due to gas loss, random noise, or some other process. However, we can take advantage of the fact that we have both  $\delta^{15}N$  and  $\delta^{40}Ar$  data, and that both  $\delta^{15}N$  and  $\delta^{40}Ar$  are controlled by identical processes, apart from  $\delta^{40}Ar$  is also affected by gas loss but  $\delta^{15}N$  is not. Therefore, the fact that we find  $\Delta\delta^{40}Ar$  is correlated with  $\Delta\delta Ar/N_2$  and  $\Delta\delta^{15}N$  is not gives us good confidence that we are detecting a true gas loss signal. This observation is very difficult to explain without invoking gas loss as the process responsible.

You are right that it is likely that the slope we calculate is not a perfect estimate of the gas loss coefficient due to uncertainty in our data and the subjectivity of our spline fitting. However, the magnitude of the gas loss correction is small and does not change our interpretation of the data for this paper. Furthermore, we believe that this approach to making a gas loss correction is likely to be of use to the community, especially as measurements of  $\delta^{40}$ Ar become more widely available in the near future. In our opinion, this justifies its inclusion in the manuscript.

Finally, the best estimate of the uncertainty in  $\delta Ar/N_2$  relevant here is the reproducibility of repeat measurements of La Jolla Air. We present this quantity in Table 1 and its value is 0.05‰ (= 12\*0.0042‰). This is smaller than the data markers in Fig. S2 (c) and (d). Note that it would be misleading to use the much larger reproducibility of the ice measurements (0.529‰) as this includes real variability between the replicates, as described on Lines 192-195 of the original manuscript.

**Line 125: shallowest? Correct here, it corresponds to the deepest firn depths!**

Apologies, this sentence was confusingly worded. We use borehole temperature data from 118-123 m depth. You are correct that this is the deepest part of the firn, but it is also the shallowest part of the Hawley dataset. We will rewrite as below.

We compute an estimate of  $\Delta T_z$  by calculating a mean temperature for the shallow and deep firn and taking the difference. For the shallow firn we use the mean of data measured between 6 and 20 m in the Giovinetto, Stevens, and Severinghaus datasets. For the deep firn, we use the mean of borehole temperatures between 118 and 123 m depth from the Hawley dataset. This gives  $\Delta T_z$  equal to 0.4°C.

---

## Author Comment (AC2)

**Reply to Referee 2**

Thank you for your constructive and detailed review of the manuscript. Our response to your comments and the changes we plan to make to the manuscript are annotated below. The text from the review comments is in black italic text, our responses are in blue, and the changes we intend to make to the manuscript are in red.

Anonymous Referee #2
Referee comment on "Gas isotope thermometry in the South Pole and Dome Fuji Ice Cores provides evidence for seasonal rectification of ice core gas records" by Jacob Davies Morgan et al., The Cryosphere Discuss., https://doi.org/10.5194/tc-2022-49-RC2, 2022

This manuscript deals with paleothermometry based on new measurements of gas isotopes in the South Pole ice core. Some measurements from the Dome F ice core are also presented. The authors are presenting a very thorough description of the methods and present improvements in the precision of d15N and d40Ar measurements which is impressive. Using firn densification modeling combined with the series of measurements of d15N and d40Ar over the South Pole ice core, they propose a reconstruction of the firn thickness and temperature gradient between the top and the bottom of the firn over the period covering 30 to 5 ka. The interpretation of the variation of the temperature gradient in the firn is not easy. Several scenarios are proposed and the authors conclude with the existence of a seasonal bias affecting the gas isotopes record.
This manuscript is well written and details the different steps of the methods and of the reasoning. It should be published within TC. I still have several comments which I think should be addressed before publications.

General comments:
I suggest to remove the whole section focused on Dome F. It is a bit disconnected from the study of the SP DTz and DCH. This section is also difficult to follow since it is not enough documented (the d15N and d40Ar data are not shown nor the origin and associated uncertainty for the DTz modelled curve). Moreover, if the Dome F data are shown, we may also wonder why we can not have the same for other sites ? showing or not a seasonal rectifier effect.

Because this comment is closely related to your final comment, we address the two together at the end of this document. We made significant changes to Section 5.2.4, including expanding our explanation of the Dome F data so that their relevance to the study is clearer to the reader.

The results displayed here raises doubts on the classical interpretation of d15N-excess (Kobashi et al., 2007; Kobashi et al., 2011) in term of surface temperature variations. A discussion revisiting these previous studies should be included here as well as clear recommendations on how to use or not the d15Nexcess for reconstructing past surface temperature variations.

We will add a paragraph to the "Broader Implications" section that discusses the relevance of our work to previous $\delta^{15}N_{excess}$ studies.

Rectification of ice core gas records has received limited attention in the literature so far, but our work argues that more careful consideration is necessary. Failure to recognise and account for rectifier effects where they do exist could potentially lead to incorrect temperature estimates. Fortunately, it is unlikely that rectifier effects would have been significant for previous gas isotope thermometry studies in Greenland (e.g., Kobashi et al., 2007, 2011; Orsi et al., 2014; Landais et al., 2004, 2006; Huber et al.,

2006). The presence of rectification via the mechanism we describe likely requires specific surface conditions such as stagnant air and a strong atmospheric temperature inversion. These conditions probably occur rarely on the Antarctic plateau and are even less common in Greenland. To have any effect on the composition of air in the deep firn and closed-off ice they must reoccur every year for many decades. Furthermore, in the case of Kobashi et al. (2011), agreement between their temperature reconstruction, regional climate model outputs and modern instrumental records also supports their analysis and interpretation. However, it might be necessary to account for rectifier effects in future gas isotope thermometry studies in Antarctica.

Comments along the manuscript:
l-25 : Precise which « temperature difference » you are speaking about.

We state on line 20 that the temperature difference is between the top and bottom of the firn column. This seems like enough detail for the abstract. The details are explained in the body of the text (section 2).

l-41: The authors were aware that the spatial temperature – isotope relationship was a surrogate for the temporal relationship and always tried to check if this was true. So I suggest to replace "thought" by "assumed"

Good point. We will rewrite the sentence as below.

Early studies calibrated $\delta^{18}O_{ice}$ using its modern-day spatial relationship with mean annual temperature near the ice core site, which was hypothesized to be identical to the relationship with temporal variations in site temperature

l-136: I do not understand why the 30 minutes delay is important for the reference gas only ? Should it not be also the case for the sample gas ?

The cooling of the bellows only affects the reference gas because it is at a higher initial pressure than the sample gas, so there is a greater decrease in pressure and therefore a greater amount of adiabatic cooling (see Section S1). However, this sentence was worded in a confusing way because the delay happens after both sample and reference gases have been admitted into the mass spectrometer. We do it this way so that both experience the delay equally. This reduces the chance that we inadvertently introduce an additional bias by treating the sample and reference gases differently. We will rewrite to include reference to the pressure difference between the sample and reference gases and to make it clear that the delay happens after both the sample and reference gases have been admitted.

The second is the inclusion of a 30-minute delay between admission of the sample and reference gas into the bellows and the beginning of the measurement sequence. This is necessary due to an initial measurement bias caused by cooling of the bellows during expansion of the reference gas, which is at a higher pressure than the sample gas prior to expansion. Both modifications are discussed in more detail in Sect. S1.

Table 1 and associated text: I am not sure how relevant it is to compare the d15N results between "Orsi Ice" and "This study SPC". Indeed, "Orsi" and "This study" obtain the same results on air measurements and the improvements mentioned in the methods section apply on both air and ice. What could explain a better precision only for ice then ?
We can thus wonder if the difference is not simply due to a poorest ice quality in "Orsi" ?
Also, it should be noted that the replicates number by "Orsi" is much larger (169) than for this study (14) which makes the comparison questionable. Can you also provide the numbers of replicate fo the LJA analyses by Orsi ? By the way, given the length of the record presented in Figure 1, I am surprised that Table 1 presents only 14 replicates. This

should be better explained. It would also be useful to give the number of replicates for Kobashi's data.

We added the number of LJA replicates from Orsi (2013) and the number or LJA and Ice replicates from Kobashi et al. (2008). Thank you for pointing out that information was missing. Orsi (2013) do not report a pooled standard deviation for LJA $\delta^{15}N_{excess}$ and Kobashi et al. (2008) does not report a pooled standard deviation for LJA or Ice $\delta^{15}N_{excess}$. We also made several other changes to the table based on comments from another referee (see below).

For this study, we were more severely limited by sample size due to the smaller diameter of SPICEcore relative to previous cores. This made it very difficult to make as many duplicate measurements as we would have liked. However, we worked hard to make sure we were able to analyze 14 duplicate samples to give us some estimate of the reproducibility.

You are correct that it is possible the improved precision is due to better ice quality for SPICEcore compared to WDC. We will add this comment to Section 4.1.

We also note smaller improvements in the reproducibility of the other measurements and that some of the improvement may be due to superior ice quality for SPICEcore.

**Table 1.** Mass normalised pooled standard deviation of replicate measurements of $\delta^{15}N$, $\delta^{40}Ar$, $\delta Ar/N_{2\ grav}$, and $\delta^{15}N_{excess}$ from either reference gas runs (REF), La Jolla air flasks (LJA), South Pole ice core samples (SPC) or other ice core samples. Units for all four isotope ratios are ‰ amu$^{-1}$ and the mass differences are 1, 4, 12, and 1 amu respectively. The final column indicates $n$, the number of samples used in the calculation.

| | $\delta^{15}N$ | $\delta^{40}Ar$ | $\delta Ar/N_{2\ grav}$ | $\delta^{15}N_{excess}$ | Num. Replicates |
|---|---|---|---|---|---|
| **This Study Ref** | 0.0020 | 0.0023 | 0.0080 | 0.0023 | 58 |
| **This Study LJA** | 0.0027 | 0.0024 | 0.0042 | 0.0019 | 40 |
| **This Study SPC** | 0.0022 | 0.0030 | 0.0432 | 0.0013 | 14 |
| **Orsi LJA** | 0.003 | 0.0025 | 0.0073 | | 10 |
| **Orsi Ice** | 0.005 | 0.0036 | 0.0331 | 0.0042 | 169 |
| **Kobashi LJA** | 0.004 | 0.0035 | 0.0114 | | |
| **Kobashi Ice** | 0.004 | 0.0040 | 0.0442 | | |

Section 5.2 (first paragraph): The arguments developed here are a bit complicated to follow after the previous section where you explained that DCH is controlled by accumulation rate itself influenced by topography. And here, you say that we expect a link between accumulation rate and temperature. I thus suggest to rewrite this paragraph so that it is coherent with the findings of the previous section.

We rewrote this section to explain our logic more clearly and to link better with the previous section.

The variability in our record of $\Delta T_z$ is initially challenging to explain. We would have anticipated a positive correlation between DCH and $\Delta T_z$ since an increase in the accumulation rate ought to result in a thicker firn column and a weaker influence of geothermal heat on the temperature at the lock-in depth. However, although DCH and $\Delta T_z$ both increase over the course of the deglaciation, we instead observe a negative correlation between DCH and $\Delta T_z$ throughout most of the record (Figure 2). There must be some other mechanism that links variability in $\Delta T_z$ to either changes in accumulation or the local topography.

Section 5.2.1 – you mention that you are using the Dage to make the reconstruction of temperature and accumulation rate but the Dage model – data fit is not shown (nor any Dage data) and it is thus difficult to follow this discussion. Moreover, when looking at the

DTz (REF), it seems that the shape of the record does mainly depend on the d18Oice – can you explain better this reconstruction of temperature ? It is important to show which data are used to constrain the shape of the temperature evolution when DECOUPLE and REF disagree. I also expect that the shape of the Kahle reconstruction is mainly imposed by the water isotopes so actually it is expected that both Kahle and REF scenarios have the same shape. This resemblance should not be taken as a strong argument to discard the surface temperature influence on the DTz scenario.

You are right to point out that the Kahle and REF scenarios are not completely independent. Both use the empirical Δage data as a constraint. The Kahle reconstruction combines this with the diffusion length proxy to calibrate the water isotope record, so its shape is very much set by the shape of the water isotopes.
The REF and DECOUPLE scenarios both use the water isotope record as an initial estimate of the temperature. However, the final shape of the curve is not dictated by the initial estimate, but by the observational constraints applied during the optimization. This is clear from comparing the REF and DECOUPLE runs, which both use the same d18O data as the initial temperature template yet look completely different after optimization due to the different constraints. For the REF fun, the constraints are DCH and Δage. For the DECOUPLE run, the constraint is our $\Delta T_z$ reconstruction. Thus, even though both experiments start with the same initial estimate, the model can produce very different final temperature histories. Therefore, the resemblance between the Kahle reconstruction and the REF scenario comes partially from the fact that they both use empirical Δage as a constraint, and partially from the fact that the other constraints (DCH and diffusion length) are in good agreement with one another. We will rewrite the paragraph to explain this caveat more clearly.

Also shown for comparison is the optimal temperature history from the REF run (Buizert et al., 2021) and a temperature history from Kahle et al. (2021) based on a calibration of δ18Oice using the SPICEcore Δage data and the diffusion length of water isotopes in the firn. Note that both temperature histories are partially constrained by the Δage data, so they are not wholly independent.

Figure 6 – is there a way to add the DTz data so that the reader sees immediately that there is a mismatch

We tried a few different ways of overlaying the data or comparing the rate of change between the data and the modelled response to a change in GHF. However, we feel that adding extra lines or shading would make this plot more difficult to interpret. It is already a busy figure, so we think it is best to not add anything else.

l-424: the mechanism is not clearly explained – this part should be rewritten.

We expanded the explanation of the convection parameterization based on your comment and comments from the other referee. The re-written paragraph is included here and also in the response below this one.

In the model run without convection, the gases diffuse towards gravitational and thermal equilibrium as they are slowly advected downwards with the densifying firn and occluded in bubbles in the lock-in zone. Because the model is one-dimensional, it is not possible to explicitly simulate a three-dimensional Rayleigh-Bénard convection cell. Instead, we model just the sinking core of a convection cell, which we parameterise as an 8 cm d$^{-1}$ downward transport of gas between 0 and 20 m. Between 20 and 25 m, the downward transport decays to zero, resulting in mass convergence that would be balanced in the real world by horizontal transport and a return flux of gas to the surface. This approach allows us to approximate how the gas isotopes respond to convection using a one-dimensional model.

- Figure 7: I am confused since the different data seem not 100% coherent with the provided explanations so probably more explanations are needed. If there is a temperature rectifier effect as suggested by the mismatch between model and data on the top 16 m, we expect a difference between d15N and d40Ar at the bottom of the firn which would then lead to a d15Nexcess signal due to seasonal rectification. Here, we see a difference but at 16m depth. Moreover, the d15Nexcess profile shown on the figures 1 and 3 does not show any 15Nexcess signal in the bubbles for the recent period, suggesting no difference between d15N and d40Ar at he bottom of present-day firn. Could the authors then better explain how they link their observation on the firn and the obesrvations in the air bubbles.

Thank you for pointing out some of the inconsistencies in the explanation of this figure. We have restructured the argument, expanded the explanation of the relevance of this figure to our argument, and added an additional figure that shows the rectifier in the deep firn. Hopefully it is clearer now, although we have not included the full re-written section here in order to keep this response somewhat concise. Briefly, we intend for Figure 7 to provide evidence that air convection in firn can advect thermal isotope signals deeper into the firn than diffusion alone can. Because the wintertime observations of $\delta^{15}$N and $\delta^{40}$Ar were made only between 0 and 16 m, it is possible they are the result of a short-lived convection event lasting only a few days immediately prior to sampling. We would only expect a rectifier signal in the deep firn if the convection persisted for several weeks/months and unfortunately we do not have any isotope data from the deep firn to test whether or not this was the case at South Pole during this sampling campaign. Instead, to demonstrate that convection in the upper firn can affect gas isotope signals in the deep firn, we performed an additional experiment under idealized conditions and included it as an additional figure. We include the new figure and the description from the text below. We also include an analogous figure for the summer rectifier in Section 5.2.4.2.
Finally, the absence of a rectifier in our youngest SPICEcore samples is not necessarily in conflict with the existence of a rectifier in the firn air dataset from South Pole. The youngest sample we analyzed is from 5 kyr BP and our SPICEcore record contains plenty of spatiotemporal variability in $\Delta T_z$ (and therefore rectifier strength) on this sort of timescale. Similar to Dome F, recent anthropogenic warming at South Pole may have helped to erase any rectification that existed in the pre-industrial era.

Sturm and Johnson (1991) demonstrated that buoyancy-driven overturning occurs readily in sub-Arctic snow in Alaska. By making hourly observations of the three-dimensional temperature field within the winter snowpack for three years, they were able to observe large horizontal temperature gradients within the snow that were initiated and maintained by columns of rising warm air and sinking cold air. This convection occurred almost continuously throughout two successive winters. There is also ample evidence for air circulation within snow and firn from Antarctica, particularly if vertical cracks allow for fast upward return flow (Giovinetto, 1963; Albert et al., 2004; Fahnestock et al., 2004; Courville et al., 2007; Severinghaus et al., 2010). Unfortunately, direct observations of changes in firn air composition associated with convection are scant since firn air sampling happens almost exclusively in the summer. However, there are published data from a winter firn air sampling campaign at South Pole. In this case, the authors did indeed find that the peak wintertime isotope signal occurred deeper than their firn air model predicted and speculated that this could be due to downward transport of the isotope anomaly by slowly sinking air (Severinghaus et al., 2001). If correct, this would provide confirmation not only of wintertime convection at South Pole, but also that thermal isotope signals can be carried down into the firn by convection without being destroyed by turbulent mixing.
To test their hypothesis, we compare their wintertime firn air measurements from South Pole with values predicted by firn air model runs with and without parameterized Rayleigh-Bénard convection (Figure 7). In the model run without convection, the gases diffuse towards gravitational and thermal equilibrium as they are slowly advected downwards with the densifying firn and occluded in bubbles in the lock-in zone. Because the model is one-dimensional, it is not possible to explicitly simulate a three-dimensional

Rayleigh-Bénard convection cell. Instead, we model just the sinking core of a convection cell, which we parameterise as an 8 cm d$^{-1}$ downward transport of gas between 0 and 20 m. Between 20 and 25 m, the downward transport decays to zero, resulting in mass convergence that would be balanced in the real world by horizontal transport and a return flux of gas to the surface. This approach allows us to approximate how the gas isotopes respond to convection using a one-dimensional model. The model run with downward transport better agrees with the observed wintertime firn air isotope ratios, with the negative wintertime values occurring deeper in the firn than in the model run with no downward advection. The model and the data therefore support our hypothesis that convection can carry seasonal thermal isotope signals down into the firn.

Because isotope data are only available in the top 16 m of the firn, we do not have an observational constraint on the strength of rectification in the deep firn, where ice core signals are recorded. To demonstrate that seasonal convection can affect isotope values in the deep firn, we perform an additional experiment with the firn air model. We simulate the isotope values in the full firn column under idealized South Pole like conditions (110 m thick firn, -51°C annual mean temperature, 7 cm a$^{-1}$ accumulation) and impose a 14 cm d$^{-1}$ downward advection throughout winter (April–September). In the model, the wintertime signal is advected deeper than the summer signal so is not fully cancelled out. This results in a -0.008‰ bias in the annual-mean signal in the deep firn compared to the control run with no downward advection (Figure 8). The bias is of comparable magnitude to the signals in our SPICEcore record, demonstrating that this mechanism could plausibly explain some of the millennial variability we observe.

[Figure]

**Figure 8.** Results of idealized modelling experiment. Panels **(a)** and **(b)** show the temperature and advection forcing applied to the firn air model. The solid lines correspond to the "with rectifier" run and the dotted line in (b) corresponds to the "without rectifier" run with no vertical advection. Panels **(c)** and **(d)** show the vertical profile of $\delta^{15}N_{excess}$ in the firn column at the end of summer and winter respectively. The grey line is the run without advection, the green line is with advection. The days corresponding to the profiles are indicated by the vertical lines in the upper panels.

l-19 and 20: The addition of the Dome F data are confusing and not helpful in this manuscript. It is a different site (much lower temperature). We have many details on the technique for measuring d15N and d40Ar but the data are not show (only firn data and only in the supplement), only DTz from d15N – d40Ar and the DTz from Buizert method but without much explanation on how it is calculated (from which data, with what kind of uncertainties ?). I suggest removing this section which does not add anything to the manuscript. On opposite, figure 7 can be helpful (but need to be shown over the whole firn depth) and is adapted to this study focused on South Pole (see however previous comment).

It seems we did not do a good enough job at explaining the relevance of the Dome F data to the paper. We chose to include the Dome F data as additional evidence that ice core gas records can be affected by rectification. The existence of rectification in multiple ice cores strengthens our argument that these effects cannot be overlooked and that more work is needed to understand where and when rectification is important for ice core gas records. As you point out, Dome F is a very different site to South Pole (colder, higher elevation, dome vs flank site). In our minds, this shows that rectification may be possible over a wide range of site characteristics on the Antarctic plateau.

In order to make this clearer to the reader and to address some of the other concerns you raised, we made the following changes:

- Restructure Section 5.4.2.1, including making the changes to the discussion of Figure 7, as described in our response above and expanding the explanation of the relevance of Dome F to the study
- Expand the Dome Fuji figure to include the Dome F $\delta^{15}$N and $\delta^{40}$Ar data, the calculated $\delta^{15}$N$_{excess}$ and $\Delta T_z$.
- Expand explanation about the Buizert et al. (2021) modelled $\Delta T_z$ and add an estimate of the associated error
- Add two new figures showing summer and winter rectification in the deep firn in an idealized firn model (see above).

The paragraphs with the most relevant and significant changes to the text are pasted below, together with the revised version of the Dome Fuji figure.

[revised manuscript text omitted]